# STAR AND FLAIR: STABILIZING AND ENRICHING RANDOMIZED NEURAL NETWORKS

## ABSTRACT

Randomized neural networks (RdNNs) surpass conventional deep models in efficiency by freezing randomly initialized input-to-hidden weights, which permits a closed-form output-layer solution and eliminates the need for backpropagation. However, they often suffer from instability and limited representation quality due to unregulated weight initialization and fixed, non-adaptive hidden mappings. Despite their widespread use, the lack of principled mechanisms to stabilize random mappings and enrich hidden representations remains largely unaddressed. To tackle these foundational issues, we introduce two novel, theoretically grounded frameworks, marking the first attempt to stabilize and enrich RdNN representations. First, **StaR** (**Sta**ble **R**epresentations) mechanism that regulates the spectrum of input-to-hidden random weight matrix by constraining singular values to a bounded interval, yielding well-conditioned hidden mappings that curb noise amplification and feature suppression. Second, **FLAiR** (**F**ew-step **L**earning for **A**daptive **I**nitialization and **R**epresentation) mechanism that applies a small, fixed number of gradient steps to input-to-hidden weights before freezing them, lightly adapting nonlinear features to task structure without incurring full backpropagation costs. We evaluate both frameworks on 146 diverse benchmark datasets, covering both binary and multiclass classification tasks using standard shallow and deep RdNN architectures. Extensive empirical results demonstrate that our methods significantly improve accuracy, stability, and generalization, while preserving the efficiency of RdNNs. Furthermore, we provide theoretical guarantees showing that **StaR** yields bounded spectral norms and well-conditioned hidden-layer transformations, and that **FLAiR** enhances representation quality through limited adaptation. Codes for baseline and **StaR/FLAiR**-enhanced models are provided in the supplementary file.

## 1 INTRODUCTION

The remarkable success of deep learning (DL) models in diverse domains, ranging from image recognition Voulodimos et al. (2018); Chen et al. (2019); Liu et al. (2024) to natural language processing Otter et al. (2021); Lauriola et al. (2022); Yin et al. (2024) stems from their ability to learn complex patterns and representations through hierarchical feature extraction LeCun et al. (2015); Luo et al. (2023). By minimizing task-specific loss functions via backpropagation and gradient-based optimization, these networks iteratively refine their parameters to learn hierarchical and expressive representations of the data. However, achieving state-of-the-art performance with deep models often requires substantial computational resources, extensive hyperparameter tuning, and long training times Goodfellow et al. (2016); Tiwari et al. (2023). Furthermore, training deep architectures can be hindered by the vanishing or exploding gradient problem Pascanu et al. (2013); Jaiswal et al. (2022); Ceni (2025).

The aforementioned limitations of DL models have prompted researchers to seek alternative neural architectures that combine predictive performance with greater computational efficiency. One promising family of approaches is randomized neural networks (RdNNs) Pao et al. (1994); Cao et al. (2018); Suganthan & Katuwal (2021); Hu et al. (2024). In contrast to conventional deep models trained by backpropagation, RdNNs randomly initialize a substantial subset of their parameters (typically the hidden-layer weights), which are then held fixed, while the remaining (often output) parameters are computed analytically in closed form. This paradigm eliminates the need for itera-

tive weight updates, substantially reducing training time and computational overhead, yet still allows RdNNs to learn complex input-output mappings Zhang & Suganthan (2016). Notably, despite their simplicity, RdNNs retain the universal approximation property, enabling them to approximate any continuous function on a compact domain arbitrarily well given sufficient hidden nodes and suitable activations Igelnik & Pao (1995); Needell et al. (2024).

In recent years, RdNNs have witnessed significant advancements, particularly in methods aimed at improving their robustness and scalability. Ensemble-based approaches have been introduced to enhance generalization and mitigate the variability Shi et al. (2021), Guo et al. (2021). Granular ball-based scalable methods have further extended the applicability of RdNNs to large-scale and high-dimensional datasets Sajid et al. (2025a). Additionally, the integration of fuzzy inference systems into RdNN frameworks has provided a way to incorporate uncertainty handling and interpretability, making them more effective for complex and imprecise decision-making tasks Sajid et al. (2024b; 2025b).

Despite several advancements, RdNNs continue to face fundamental issues stemming from fixed, randomly initialized input-to-hidden weights, which can produce poorly conditioned hidden representations, amplify input noise, and result in instability and inconsistent performance across datasets Li & Wang (2017); Scardapane & Wang (2017). Yet, no principled framework has been proposed to regulate the randomness at the heart of RdNNs. The core challenges arising from unregulated and fixed randomness—namely, ill-conditioned hidden representations and high variability in learned features—are formally articulated in Problem Statements **P1** and **P2**, which are elaborated in the subsequent section.

Although techniques such as better initialization Glorot & Bengio (2010), orthogonalization Saxe et al. (2013), and spectral normalization Miyato et al. (2018) have been proposed to mitigate instability in traditional deep networks—where weights are iteratively updated via backpropagation—these methods are tailored for gradient-based optimization and are not applicable to RdNNs, where weights remain fixed after random initialization. Within the RdNN paradigm, limited efforts have been made to address the instability and low representational quality that arise from unregulated fixed weights. Regularization techniques, such as $\ell_1$ penalties Zhang et al. (2019), and ensemble-based methods Cao et al. (2012), have been explored to improve generalization. However, these approaches primarily operate at the output level, leaving the root causes—namely, unstable hidden transformations and uninformative or poorly expressive hidden representations—unaddressed. To date, no systematic framework has been developed to tackle these limitations directly, leaving a critical gap in the literature.

## 2 CONTRIBUTIONS

To directly address the core challenges of instability and poor representation quality in RdNNs, we propose two novel frameworks: **StaR** (**Sta**ble **R**epresentations) and **FLAiR** (**F**ew-step **L**earning for **A**daptive **I**nitialization and **R**epresentation). These frameworks provide the first principled mechanisms to ensure numerical stability and improve representational quality in RdNNs by addressing the effects of fixed randomness.

The **StaR** framework is theoretically motivated by insights from *Random Matrix Theory* (RMT) Edelman & Rao (2005); Tao (2012), which provides a rigorous mathematical foundation for characterizing the spectral properties of large random matrices—particularly their singular value distributions. RMT has shown that unstructured random matrices, such as those used in RdNNs, often exhibit extreme singular values, leading to ill-conditioned transformations that amplify noise or suppress informative signals. Drawing on this perspective, **StaR** seeks to regulate the spectrum of random input-to-hidden weight matrices, ensuring that the resulting representations are stable and well-conditioned. This spectral regularization strategy directly addresses the core instability and numerical conditioning issues, while preserving the computational simplicity of RdNNs.

The **FLAiR** framework is inspired by recent theoretical insights into representation learning in high-dimensional settings. Specifically, the work of Ba et al. Ba et al. (2022) reveals that even a single gradient step can lead to a considerable advantage over random features. While this phenomenon has been studied in iterative training regimes, its implications are particularly valuable for RdNNs, where the input-to-hidden weights are fixed and untrained. **FLAiR** harnesses this insight by in-

troducing a lightweight warm-up phase, wherein a small number of gradient steps are applied to the input-to-hidden weights before freezing. This early-stage adaptation promotes task-aware feature extraction and improves the representational capacity of RdNNs without compromising their hallmark efficiency. By doing so, **FLAiR** enables RdNNs to move beyond purely random transformations, offering a principled means to inject data-dependent structure into otherwise fixed feature maps.

Together, **StaR** and **FLAiR** mark the first effort to systematically enhance both the stability and expressiveness of RdNNs—directly at their structural core. By regulating spectral properties and introducing lightweight data-driven adaptation, these methods retain the efficiency of RdNNs while systematically resolving long-standing structural limitations. Notably, both frameworks are architecture-agnostic and can be integrated into any RdNN variant, paving the way for more stable, expressive, and theoretically grounded randomized models.

## 3 PRELIMINARIES AND PROBLEM DEFINITION

**RdNN setting.** Let $X \in \mathbb{R}^{m \times d}$ denote the input data matrix with $m$ samples and $d$ features, and let $Y \in \mathbb{R}^{m \times n}$ be the target matrix (e.g., one-hot encoded labels for classification). A RdNN samples a hidden-layer weight matrix $W \in \mathbb{R}^{d \times h}$ from a fixed distribution (typically $\mathcal{U}[-1, 1]$ or $\mathcal{N}(0, 1)$) and freezes it throughout training. The hidden-layer output is given by

$$H = \phi(XW + B) \in \mathbb{R}^{m \times h}, \tag{1}$$

where $\phi : \mathbb{R} \to \mathbb{R}$ is a nonlinear activation function applied elementwise, and $B \in \mathbb{R}^{m \times h}$ is a bias matrix. The output weights $\Theta \in \mathbb{R}^{h \times n}$ are computed in closed form via regularized least squares:

$$\Theta^\star(W) = \arg\min_\Theta \ \|H\Theta - Y\|_F^2 + \lambda \|\Theta\|_F^2$$

$$= (H^\top H + \lambda I_h)^{-1} H^\top Y, \tag{2}$$

where $\lambda \geq 0$ is a regularization parameter. The above formulation captures the core learning structure shared by a wide class of RdNN models, where training reduces to solving a linear system after a fixed random feature transformation. Detailed formulation and architecture of standard RdNN models are discussed in Section A of the Appendix.

**Spectral preliminaries.** The stability and expressiveness of RdNNs critically depend on the spectral properties of the random weight matrix $W$. Let $W = U \cdot \Sigma \cdot V^\top$ be the singular value decomposition (SVD) of $W$, where $\Sigma = \text{diag}(\sigma_1, \ldots, \sigma_r)$ contains the singular values of $W$, and $r = \min(d, h)$. The spectral norm $\|W\|_2 = \sigma_{\max}$ determines the maximum amplification of input directions, while the smallest singular value $\sigma_{\min}$ controls the degree of contraction. The condition number $\kappa(W) = \sigma_{\max}/\sigma_{\min}$ quantifies the sensitivity of matrix inversion to perturbations and is a standard measure of numerical stability Golub & Van Loan (2013). An ill-conditioned $W$, characterized by a large condition number, can lead to unstable or poorly scaled hidden-layer representations $H$ Zhang et al. (2018). These problems amplify noise, suppress informative features, and degrade the numerical stability of the closed-form solution in equation 2. Moreover, excessive variation in spectral norms across initializations contributes to inconsistent performance.

PROBLEM STATEMENT

**P1: Spectral Instability and Poor Conditioning.** In RdNNs, the hidden-layer transformation $H = \phi(XW + B)$ is governed by the spectral structure of the randomly initialized matrix $W$. Unregulated $W$ may exhibit heavy-tailed or degenerate singular spectra, causing (i) noise amplification (large $\sigma_{\max}$), (ii) feature suppression (small $\sigma_{\min}$), and (iii) numerical instability in solving equation 2 (large $\kappa(W)$). These issues can severely degrade the stability and robustness of RdNNs. The core objective is thus to *stabilize* the hidden-layer transformation by enforcing a controlled and well-conditioned singular spectrum in $W$.

**P2: Inflexible Representations from Frozen Random Weights.** While fixing the random weight matrix $W$ allows RdNNs to avoid iterative backpropagation and enjoy efficient closed-form training, it inherently limits the expressiveness and task-specific alignment of the hidden transformation

$H = \phi(XW + B)$. Since $W$ is sampled randomly and remains frozen, the resulting feature representations may fail to learn discriminative patterns essential for the downstream task. Additionally, the randomness of $W$ induces high variability in performance across different initializations, compromising model reliability. The fundamental challenge is thus to enrich the representational quality of $H$ by enabling limited, computationally light adaptation of $W$, without sacrificing the efficiency of closed-form training.

## 4 METHOD

To address the limitations identified in **P1** (spectral instability) and **P2** (inflexible representations), we propose two frameworks:

- **StaR** (*Stable Representations*) directly targets **P1** by enforcing spectral regularity in the random weight matrix $W$. It projects the singular values of $W$ onto a prescribed bounded interval, yielding a well-conditioned transformation $H$ that mitigates both noise amplification and feature suppression.

- **FLAiR** (*Few-step Learning for Adaptive Initialization and Representation*) addresses **P2** by introducing a lightweight adaptation stage. A small number of gradient updates refine $W$ before freezing, enabling task-aware hidden-layer representations without sacrificing closed-form output-layer training.

While **StaR** and **FLAiR** are designed to address **P1** and **P2** respectively, both frameworks contribute to improving the hidden transformation $H = \phi(XW + B)$ through distinct mechanisms. We now describe each framework in detail.

### 4.1 STAR: STABLE REPRESENTATIONS

**StaR** is a principled singular value regulation mechanism designed to mitigate spectral pathologies in RdNNs by projecting the spectrum of the randomly initialized weight matrix $W \in \mathbb{R}^{d \times h}$ onto a bounded interval. This ensures that the resulting hidden-layer transformation $H = \phi(XW + B)$ remains well-conditioned, thereby addressing **P1**. The detailed steps of the **StaR** framework are as follows:

1. **Singular value decomposition:** The randomly initialized weight matrix $W \in \mathbb{R}^{d \times h}$ is decomposed as:
$$W = U \cdot \Sigma \cdot V^\top,$$
   where $U \in \mathbb{R}^{d \times d}$ and $V \in \mathbb{R}^{h \times h}$ are orthogonal matrices containing the left and right singular vectors of $W$, and $\Sigma = \mathrm{diag}(\sigma_1, \sigma_2, \ldots, \sigma_r)$ is a diagonal matrix of singular values with $\sigma_i \geq 0$.

2. **Singular value rescaling:** To improve the conditioning of the weight matrix and ensure stable transformations, the singular values in $\Sigma$ are linearly rescaled to lie within a target interval $[\sigma_{\text{low}}, \ \sigma_{\text{high}}]$, where $\sigma_{\text{low}} > 0$. This guarantees that the singular values of the regulated matrix $W'$ satisfy:
$$\sigma_{\text{low}} \leq \sigma'_i \leq \sigma_{\text{high}}, \quad \forall i \in \{1, 2, \ldots, r\}. \tag{3}$$

   The rescaled singular values are computed as:
$$\sigma'_i = \frac{\sigma_i - \sigma_{\text{min}}}{\sigma_{\text{max}} - \sigma_{\text{min}} + \varepsilon} \cdot (\sigma_{\text{high}} - \sigma_{\text{low}}) + \sigma_{\text{low}}, \ \forall i, \tag{4}$$

   where $\sigma_{\text{min}} = \min_i \sigma_i$, $\sigma_{\text{max}} = \max_i \sigma_i$, and $\varepsilon$ is a small constant added for numerical stability.

   **Conditioning Guarantee:** As a result of this bounded rescaling, the condition number of the regulated matrix is explicitly controlled:
$$\kappa(W') = \frac{\sigma_{\text{high}}}{\sigma_{\text{low}}}, \tag{5}$$

---

**Algorithm 1** $\texttt{StaR}$: **Sta**ble **R**epresentation via Spectral Rescaling

---

**Input:** Random weight matrix $W \in \mathbb{R}^{d \times h}$; spectral bounds $\sigma_{\text{low}} > 0$, $\sigma_{\text{high}} > \sigma_{\text{low}}$

**Output:** Regulated matrix $W' \in \mathbb{R}^{d \times h}$

**1. Compute SVD:** $W = U \cdot \Sigma \cdot V^\top$

**2. Extract singular values:** $\Sigma = \text{diag}(\sigma_1, \ldots, \sigma_r)$

**3. Normalize and Rescale:**

$$\sigma_{\min} \leftarrow \min_i \sigma_i, \quad \sigma_{\max} \leftarrow \max_i \sigma_i$$

$$\sigma_i' \leftarrow \frac{\sigma_i - \sigma_{\min}}{\sigma_{\max} - \sigma_{\min} + \varepsilon} \cdot (\sigma_{\text{high}} - \sigma_{\text{low}}) + \sigma_{\text{low}}, \quad \forall i$$

$$\Sigma' \leftarrow \text{diag}(\sigma_1', \ldots, \sigma_r')$$

**4. Reconstruct:** $W' = U \cdot \Sigma' \cdot V^\top$

**Return:** $W'$

---

    which prevents ill-conditioning and ensures well-behaved hidden-layer transformations. By avoiding extremely small singular values, $\texttt{StaR}$ reduces the risk of feature collapse and information loss, while bounding large singular values mitigates noise amplification and instability.

3. **Reconstruction of the regulated weight matrix:** The regulated weight matrix $W'$ is reconstructed using the rescaled singular values:

$$W' = U \cdot \Sigma' \cdot V^\top, \tag{6}$$

where $\Sigma' = \text{diag}(\sigma_1', \sigma_2', \ldots, \sigma_r')$. This procedure preserves the directional geometry encoded by the singular vectors while correcting the pathological scaling in the singular values. The resulting matrix $W'$ ensures stable and well-conditioned hidden transformations, prevents noise amplification and feature suppression.

Algorithm 1 summarizes the $\texttt{StaR}$ procedure.

To rigorously characterize the benefits of spectral regulation, we formalize two key advantages of $\texttt{StaR}$ in terms of hidden-layer conditioning and stability. Specifically, Lemma 1 (see Appendix B) establishes (i)—that $\texttt{StaR}$ controls the condition number of the random weight matrix—while Lemma 2 (see Appendix B) establishes (ii)—that $\texttt{StaR}$ induces Lipschitz-continuous transformations, ensuring numerically stable and bounded sensitivity to input perturbations.

These theoretical results formally support the conditioning and stability improvements introduced by $\texttt{StaR}$.

### 4.2 $\texttt{FLAiR}$: FEW-STEP LEARNING FOR ADAPTIVE INITIALIZATION AND REPRESENTATION

$\texttt{FLAiR}$ is a general framework for enhancing the representation capacity of RdNN by introducing a short, few-step learning phase for input-to-hidden weights. Instead of relying solely on fixed random projections, $\texttt{FLAiR}$ introduces a short warm-up phase wherein the input-to-hidden weights (and biases) are updated using gradient-based optimization for a few epochs. These refined weights are then frozen, and the output weights are subsequently computed in closed-form. This two-stage procedure improves feature alignment, reduces sensitivity to random initialization, and enhances downstream performance, thereby addressing **P2**. The $\texttt{FLAiR}$ mechanism comprises the following steps:

1. **Initialization:** Random weights $W$ and biases $B$ are sampled, as per the standard RdNN setup.

2. **Warm-up Training (Few-step Adaptive Update):** To improve representation quality, $\texttt{FLAiR}$ introduces a short warm-up phase that adaptively updates the hidden parameters $W$ and $B$ for a small number of steps $T$ using gradient descent (e.g., Adam optimizer). At

---

**Algorithm 2 `FLAiR`: F**ew-step **L**earning for **A**daptive **I**nitialization and **R**epresentation

---

**Input:** Data matrix $X \in \mathbb{R}^{m \times d}$, target matrix $Y \in \mathbb{R}^{m \times n}$; number of hidden units $h$; activation $\phi(\cdot)$; warm-up epochs $T$; learning rate $\eta$; regularization $\lambda > 0$

**Output:** Output weights $\Theta \in \mathbb{R}^{h \times n}$; refined weights $W^{(T)}, B^{(T)}$

**1. Random Initialization:** Sample $W^{(0)} \in \mathbb{R}^{d \times h}$, $B^{(0)} \in \mathbb{R}^{1 \times h}$

**for** $t = 1$ to $T$ **do**

  **a. Hidden Representation:** $H^{(t)} = \phi(XW^{(t-1)} + \mathbf{1}B^{(t-1)})$

  **b. Closed-form Output Estimation:**

$$\Theta^{(t)} = (H^{(t)\top}H^{(t)} + \lambda I_h)^{-1}H^{(t)\top}Y$$

  **c. Prediction:** $\hat{Y}^{(t)} = H^{(t)}\Theta^{(t)}$

  **d. Loss Computation:** $\mathcal{L}^{(t)} = \|\hat{Y}^{(t)} - Y\|_F^2$

  **e. Backpropagation:** Update $W^{(t-1)}, B^{(t-1)}$ using gradients from $\mathcal{L}^{(t)}$ (e.g., via Adam)

**end for**

**2. Freeze Parameters:** $W^{(T)} \leftarrow W^{(T-1)}$, $B^{(T)} \leftarrow B^{(T-1)}$

**3. Final Hidden Representation:** $H = \phi(XW^{(T)} + \mathbf{1}B^{(T)})$

**4. Final Output Learning:**

$$\Theta = (H^\top H + \lambda I_h)^{-1}H^\top Y$$

**Return:** $\Theta$, $W^{(T)}$, $B^{(T)}$

---

each warm-up epoch $t = 1, \ldots, T$, the current hidden features are computed as:

$$H^{(t)} = \phi(XW^{(t)} + B^{(t)}) \in \mathbb{R}^{m \times h}, \tag{7}$$

where $W^{(t)}$ and $B^{(t)}$ are the current hidden weights and bias matrix.

To provide supervision for gradient updates, a temporary output weight matrix $\Theta^{(t)}$ is computed in closed form by solving the ridge regression:

$$\Theta^{(t)} = \arg\min_{\Theta} \|H^{(t)}\Theta - Y\|_F^2 + \lambda\|\Theta\|_F^2. \tag{8}$$

The prediction $\hat{Y}^{(t)} = H^{(t)}\Theta^{(t)}$ is used to evaluate a squared error loss:

$$\mathcal{L}^{(t)} = \|\hat{Y}^{(t)} - Y\|_F^2, \tag{9}$$

which is backpropagated only through $W^{(t)}$ and $B^{(t)}$. This improves the alignment of the random features with the target signal, while keeping the training efficient since no backpropagation is applied to $\Theta$. After $T$ warm-up steps, the refined hidden weights are frozen, and final output weights are recomputed in closed form using the final hidden activations $H^{(T)}$.

3. **Closed-form Output Layer Learning:** Once the warm-up phase concludes, the updated hidden parameters $W^{(T)}$ and $B^{(T)}$ are frozen. The final hidden representation is computed as:

$$H = \phi(XW^{(T)} + B^{(T)}). \tag{10}$$

Then, the output weights $\Theta \in \mathbb{R}^{h \times n}$ are computed in closed form using regularized least squares:

$$\Theta = (H^\top H + \lambda I_h)^{-1}H^\top Y, \tag{11}$$

where $I_h$ is the $h \times h$ identity matrix. This step is efficient and avoids iterative training, preserving the computational simplicity of RdNNs while benefiting from the improved feature representations produced during warm-up.

Algorithm 2 provides a detailed summary of the `FLAiR` framework, outlining the full warm-up and closed-form training procedure for input-to-hidden adaptation in RdNNs.

*Remark* 1. While the provided procedure presents **FLAiR** in the context of a shallow RdNN with a single hidden layer for notational clarity, the same principle extends seamlessly to deeper or structured RdNN architectures by applying the warm-up and freeze mechanism to each layer or subcomponent independently.

We further provide a formal result showing that the few-step supervised updates to the hidden parameters yield a monotonic *reduction in prediction loss* over warm-up epochs. This justifies the effectiveness of **FLAiR**'s adaptive initialization in aligning random features with the target task; see Lemma 3 in Appendix B. This result establishes a theoretical justification for **FLAiR**'s supervised warm-up strategy by demonstrating its guaranteed reduction in prediction error across gradient steps, thereby validating its role in improving hidden-layer alignment.

## 5 EMPIRICAL RESULTS

This section rigorously evaluates the effectiveness of the proposed **StaR** and **FLAiR** frameworks by integrating them into a diverse suite of representative RdNN architectures, encompassing shallow, multi-layer, and deep variant. Specifically, we consider Random Vector Functional Link Network (RVFL) Pao et al. (1994), Extreme Learning Machine (ELM) Huang et al. (2006), Broad Learning System (BLS) Chen & Liu (2017), and Deep Random Vector Functional Link Network (dRVFL) Shi et al. (2021). To assess the impact of proposed frameworks, we instantiate two distinct families of enhanced models:

- **StaR-enhanced RdNNs**: These variants apply the spectral regularization procedure of **StaR** to the randomly initialized weight matrices. The resulting models are denoted as **StaR**-RVFL, **StaR**-ELM, **StaR**-BLS, and **StaR**-dRVFL.

- **FLAiR-enhanced RdNNs**: These models incorporate the few-step adaptive initialization strategy of **FLAiR**. The corresponding enhanced variants are denoted as **FLAiR**-RVFL, **FLAiR**-ELM, **FLAiR**-BLS, and **FLAiR**-dRVFL.

The performance of these models is benchmarked against their baseline counterparts using a suite of statistical metrics and tests, including accuracy, standard deviation, rank analysis, Friedman test, Nemenyi posthoc test, and win-tie-loss analysis, ensuring a comprehensive and reliable analysis. The dataset description and detailed experimental setup is provided in Section C of the Appendix.

**Dataset Description:** The evaluation is conducted across 146 binary and multiclass benchmark datasets sourced from the UCI Dua & Graff (2017) and KEEL Derrac et al. (2015) repositories, ensuring robust generalization and diversity. Among the datasets, 93 are binary classification problems, and 53 are multiclass classification problems, with the number of classes ranging from 2 to 26. The number of samples spans from 16 to 130,064, and the number of features varies from 2 to 262, covering both small-scale and large-scale datasets as well as low-dimensional and high-dimensional data. For shallow (RVFL, ELM) and broad (BLS) architectures, along with their enhanced variants, all 146 datasets are utilized. In contrast, due to the higher computational complexity associated with training deep randomized models such as dRVFL and its enhanced counterparts, evaluation is conducted on a representative subset of 71 datasets (36 binary and 35 multiclass). The details of binary and multiclass datasets are provided in Section C of the Appendix.

### 5.1 PERFORMANCE EVALUATION AND ANALYSIS

We now present a detailed quantitative evaluation of the proposed **StaR** and **FLAiR** frameworks across 146 binary and multiclass classification datasets from various domains.

### 5.1.1 EFFECTIVENESS OF **StaR** FRAMEWORK

Tables 1a and 1b present the average accuracy, standard deviation, and rank statistics for the **StaR**-enhanced models (**StaR**-RVFL, **StaR**-ELM, **StaR**-BLS, and **StaR**-dRVFL) compared to their respective baselines on binary and multiclass datasets, respectively. The detailed results on each dataset are provided in the Section D of the Appendix. On binary classification datasets, for RVFL, **StaR** yields an absolute accuracy gain of 0.67% (from 85.13% to 85.80%) and reduces the standard deviation by 0.94 (from 5.87 to 4.93). In the case of ELM, **StaR** improves accuracy by 0.79% and

Table 1: Average performance comparison of baseline and **StaR**-enhanced RdNN models on (a) binary and (b) multiclass classification datasets. ↑ indicates improved accuracy (Acc.); ↓ denotes reduced standard deviation (Std.) and rank.

| Model | Acc. ↑ | Std. ↓ | Rank (Acc.) ↓ | Rank (Std.) ↓ |
|---|---|---|---|---|
| RVFL | 85.13 | 5.87 | 1.82 | 1.70 |
| **StaR**-RVFL | 85.80 ↑ | 4.93 ↓ | 1.18 ↓ | 1.30 ↓ |
| ELM | 84.98 | 5.82 | 1.77 | 1.71 |
| **StaR**-ELM | 85.77 ↑ | 4.98 ↓ | 1.23 ↓ | 1.29 ↓ |
| BLS | 85.60 | 5.31 | 1.83 | 1.69 |
| **StaR**-BLS | 86.04 ↑ | 4.56 ↓ | 1.17 ↓ | 1.31 ↓ |
| dRVFL | 83.13 | 6.84 | 1.97 | 1.97 |
| **StaR**-dRVFL | 84.09 ↑ | 5.34 ↓ | 1.03 ↓ | 1.03 ↓ |

(a) Binary datasets

| Model | Acc. ↑ | Std. ↓ | Rank (Acc.) ↓ | Rank (Std.) ↓ |
|---|---|---|---|---|
| RVFL | 77.29 | 8.80 | 1.82 | 1.85 |
| **StaR**-RVFL | 78.01 ↑ | 6.74 ↓ | 1.18 ↓ | 1.15 ↓ |
| ELM | 76.78 | 9.24 | 1.86 | 1.75 |
| **StaR**-ELM | 77.88 ↑ | 7.93 ↓ | 1.14 ↓ | 1.25 ↓ |
| BLS | 77.73 | 6.85 | 1.93 | 1.74 |
| **StaR**-BLS | 78.53 ↑ | 5.99 ↓ | 1.07 ↓ | 1.26 ↓ |
| dRVFL | 78.37 | 7.79 | 1.91 | 1.91 |
| **StaR**-dRVFL | 79.37 ↑ | 6.29 ↓ | 1.09 ↓ | 1.09 ↓ |

(b) Multiclass datasets

lowers variability by 0.84. BLS also benefits from **StaR**, showing a 0.44% increase in accuracy and a 0.75 reduction in standard deviation. Notably, dRVFL, despite being a deeper architecture, records the highest improvement in accuracy of 0.96%, along with a significant 1.50 decrease in standard deviation. Even greater gains are observed on multiclass classification datasets, where the benefits of integrating **StaR** become more prominent. For RVFL, the integration of **StaR** results in a notable accuracy improvement of 0.72% (from 77.29% to 78.01%) and a substantial 2.06 reduction in standard deviation, highlighting improved robustness across diverse class distributions. In the case of ELM, **StaR** provides a 1.10% boost in accuracy and a 1.31 decrease in variability, reinforcing its effectiveness even under higher output dimensionality. **StaR**-BLS achieves a 0.80% increase in accuracy and reduces the standard deviation by 0.86, showing enhanced stability. Once again, the deep architecture dRVFL benefits the most—registering the highest absolute accuracy gain of 1.00% and a 1.50 drop in standard deviation. These findings further affirm the consistent generalization and variance-reduction capabilities of **StaR**, especially in challenging multiclass scenarios. These empirical results strongly validate the theoretical foundations of the **StaR** framework. By constraining the spectral properties of random weight matrices, **StaR** enforces controlled transformations in the hidden layer, thereby mitigating noise amplification and improving stability across training runs. The observed improvements in both accuracy and standard deviation demonstrate that spectral regularization is an effective mechanism for enhancing generalization and reducing performance variability in RdNNs.

To further substantiate these findings, we analyze the average ranks based on accuracy and standard deviation, which jointly reflect generalization and stability across datasets. On binary classification tasks, all **StaR**-enhanced models exhibit lower ranks compared to their baselines. For instance, the accuracy rank of **StaR**-RVFL improves from 1.82 to 1.18, while its standard deviation rank improves from 1.70 to 1.30. Similarly, **StaR**-ELM and **StaR**-BLS reduce their accuracy ranks from 1.77 and 1.83 to 1.23 and 1.17, respectively, and their standard deviation ranks from 1.71 and 1.69 to 1.29 and 1.31. Notably, **StaR**-dRVFL achieves the best ranks in both dimensions: 1.03 for accuracy and 1.03 for standard deviation. These improvements are even more pronounced on multiclass datasets, where **StaR**-RVFL reduces its ranks from 1.82 and 1.85 to 1.18 and 1.15, respectively. Likewise, **StaR**-ELM achieves accuracy and stability ranks of 1.14 and 1.25, outperforming its baselines (1.86 and 1.75), while **StaR**-BLS improves from (1.93, 1.74) to (1.07, 1.26). Again, **StaR**-dRVFL delivers the most competitive ranking (1.09, 1.09), highlighting its effectiveness in high-dimensional, multi-class scenarios. These consistently lower ranks confirm the superior and robust performance of **StaR**-enhanced models across diverse datasets. Moreover, to statistically validate the observed improvements, we conduct multiple statistical tests, including the Friedman test, the Nemenyi posthoc test, and the win-tie-loss analysis Demšar (2006). The detailed results are provided in Section E of the Appendix.

### 5.1.2 EFFECTIVENESS OF **FLAiR** FRAMEWORK

Tables 2a and 2b summarize the average accuracy, standard deviation, and rank metrics for the **FLAiR**-enhanced models across binary and multiclass datasets. Detailed results for each dataset are included in Section *S.IV* of the supplement file. On binary classification tasks, **FLAiR**-RVFL improves accuracy by 1.44% (87.55% to 88.99%) and reduces the standard deviation by 0.79 (4.90

Table 2: Average performance comparison of baseline and **FLAiR**-enhanced RdNN models on (a) binary and (b) multiclass classification datasets. ↑ indicates improved accuracy (Acc.); ↓ denotes reduced standard deviation (Std.) and rank.

| Model | Acc. ↑ | Std. ↓ | Rank (Acc.) ↓ | Rank (Std.) ↓ |
|---|---|---|---|---|
| RVFL | 87.55 | 4.90 | 1.94 | 1.67 |
| **FLAiR**-RVFL | **88.99** ↑ | **4.11** ↓ | **1.06** ↓ | **1.33** ↓ |
| ELM | 87.49 | 4.71 | 1.95 | 1.71 |
| **FLAiR**-ELM | **89.15** ↑ | **3.68** ↓ | **1.05** ↓ | **1.29** ↓ |
| BLS | 86.75 | 3.99 | 1.96 | 1.65 |
| **FLAiR**-BLS | **88.95** ↑ | **3.39** ↓ | **1.04** ↓ | **1.35** ↓ |
| dRVFL | 83.13 | 6.84 | 1.96 | 1.90 |
| **FLAiR**-dRVFL | **85.58** ↑ | **4.49** ↓ | **1.04** ↓ | **1.10** ↓ |

(a) Binary datasets

| Model | Acc. ↑ | Std. ↓ | Rank (Acc.) ↓ | Rank (Std.) ↓ |
|---|---|---|---|---|
| RVFL | 79.47 | 6.86 | 1.99 | 1.61 |
| **FLAiR**-RVFL | **81.39** ↑ | **6.40** ↓ | **1.01** ↓ | **1.39** ↓ |
| ELM | 78.90 | 6.84 | 2.00 | 1.66 |
| **FLAiR**-ELM | **81.43** ↑ | **6.17** ↓ | **1.00** ↓ | **1.34** ↓ |
| BLS | 80.29 | 8.79 | 2.00 | 1.58 |
| **FLAiR**-BLS | **83.56** ↑ | **6.86** ↓ | **1.00** ↓ | **1.42** ↓ |
| dRVFL | 78.37 | 7.79 | 2.00 | 1.60 |
| **FLAiR**-dRVFL | **81.51** ↑ | **6.85** ↓ | **1.00** ↓ | **1.40** ↓ |

(b) Multiclass datasets

to 4.11). **FLAiR**-ELM shows an even larger gain of 1.66% in accuracy and a reduction of 1.03 in standard deviation. **FLAiR**-BLS improves accuracy by 2.20% and variability by 0.60. Notably, **FLAiR**-dRVFL records a substantial 2.45% increase in accuracy and a 2.35 drop in standard deviation, highlighting the framework's strong impact on deeper architectures. Performance gains are more pronounced in multiclass settings. **FLAiR**-RVFL improves accuracy by 1.92% and reduces standard deviation by 0.46. **FLAiR**-ELM exhibits a stronger gain, improving accuracy by 2.53% and reducing variability by 0.67. The highest accuracy improvement of 3.27% is achieved by **FLAiR**-BLS, which also shows a 1.93 decrease in standard deviation. **FLAiR**-dRVFL sees a 3.14% boost and a 0.94 drop in variability, confirming its ability to stabilize deep models in high-dimensional settings. These results empirically validate the core principle behind **FLAiR**: by briefly optimizing the input-to-hidden weights using gradient-based warm-up, **FLAiR** introduces task-specific structure into the initial random projections. This adaptive refinement aligns the hidden-layer transformations with the data distribution, thereby enhancing feature expressiveness and improves generalization performance across diverse architectures and tasks.

Rank-based evaluation further substantiates the superiority of **FLAiR**-enhanced models. On binary datasets, they achieve top accuracy ranks (1.04–1.06) and consistently lower standard deviation ranks (1.10–1.35), reflecting improved generalization and stability. On multiclass datasets, all models attain the best possible accuracy ranks (1.00–1.01), indicating that **FLAiR**-based models outperform their baselines across nearly every dataset. The corresponding drop in standard deviation ranks further confirms **FLAiR**'s ability to stabilize performance across varied classification tasks. Additional statistical significance test results (Friedman test, Nemenyi posthoc test, and win-tie-loss analysis) are provided in Section E of the Appendix.

Complementing these results, we empirically validate Lemma 3 by analyzing the progression of prediction loss over increasing warm-up epochs. As presented in Section F of the Appendix, the loss consistently exhibits a monotonic decline across multiple datasets, confirming that **FLAiR** effectively reduces prediction error through a few gradient updates of the input-to-hidden weights, thereby enhancing representational quality.

## 6 CONCLUSIONS

This work introduced two principled frameworks—**StaR** and **FLAiR**—that directly address the long-standing challenges of instability and weak representational capacity in randomized neural networks (RdNNs). **StaR** improves numerical stability by regulating the spectral characteristics of random weight matrices, while **FLAiR** enhances representational quality through lightweight, task-aware adaptation of fixed hidden-layer mappings. Rigorous theoretical analysis and comprehensive evaluations across 146 benchmark datasets validate the efficacy, stability, and generalization capabilities of both frameworks. Crucially, **StaR** and **FLAiR** are model-agnostic and can be seamlessly integrated into a wide range of RdNN architectures. We hope these contributions inspire further research for stabilizing and enriching randomized representations in modern neural learning. All implementation details, including code for baseline and **StaR/FLAiR**-enhanced models, are provided in the supplementary file.

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

APPENDIX

# A ARCHITECTURAL AND MATHEMATICAL FORMULATION OF STANDARD RdNN MODELS

## A.1 FORMULATION AND ARCHITECTURE OF RVFL AND ELM PAO ET AL. (1994); HUANG ET AL. (2006)

RVFL and ELM are single-layer feed-forward RdNNs that rely on hidden layer transformations and closed-form solutions for output weight computation. While their architectures share significant similarities, the primary distinction lies in the presence of direct input-to-output connections in RVFL, which are absent in ELM. Their common formulation is described as follows:

**Hidden Layer:** The hidden layer transforms the input data $X$ using a random weight matrix $W \in \mathbb{R}^{d \times h}$ and bias $B \in \mathbb{R}^{m \times h}$:

$$H = \phi(XW + B), \tag{12}$$

where $\phi(\cdot)$ is the non-linear function and $h$ represents the number of nodes in the hidden layer.

**Output Layer:** In RVFL, the final output combines the hidden layer outputs and the original inputs to form an augmented representation:

$$A = [X \,|\, H], \tag{13}$$

whereas in ELM, the output is directly derived from the hidden layer outputs:

$$A = H. \tag{14}$$

The output weight matrix $\Theta \in \mathbb{R}^{h \times n_{\text{class}}}$ is computed in both architectures using a closed-form solution:

$$\Theta = (A^\top A + \lambda I)^{-1} A^\top Y, \tag{15}$$

where $\lambda$ is a regularization parameter.

**Key Difference:** RVFL incorporates direct input-to-output connections ($Z = [X \,|\, H]$), which preserve input information and enhance robustness. In contrast, ELM omits this feature ($Z = H$), resulting in a simpler architecture that trades off some robustness for faster training.

## A.2 FORMULATION AND ARCHITECTURE OF BLS CHEN & LIU (2017)

BLS represents a significant advancement in RdNNs by utilizing a flat architecture instead of the deep hierarchical structures seen in traditional deep learning. The architecture of BLS consists of three primary components: the feature layer, the enhancement layer, and the output layer. The feature layer is responsible for extracting meaningful features from the input data by generating multiple groups of feature nodes through random projection and non-linear transformations. On the other hand, the enhancement layer enriches the feature representations by applying additional non-linear transformations, thereby constructing enhancement nodes that provide diversity and robustness. The outputs from both layers are concatenated to form a final matrix, which is used to compute the output weights via a closed-form solution. Its architecture can be described as follows:

**Feature Layer:** For the input matrix $X$, the feature layer generates $q$ windows of feature nodes. Each window $f_i$ contains $p$ nodes, which are computed as:

$$Z_{f_i} = \phi(XW_{f_i} + B_{f_i}), \tag{16}$$

where $W_{f_i} \in \mathbb{R}^{d \times p}$ is a random weight matrix, $B_{f_i} \in \mathbb{R}^{m \times p}$ is a bias matrix, $\phi(\cdot)$ is a non-linear function, and $Z_{f_i} \in \mathbb{R}^{m \times p}$ represents the output of the $i^{th}$ feature node window. The outputs of all feature windows are concatenated to form the overall feature layer output:

$$Z = [Z_{f_1}, Z_{f_2}, \ldots, Z_{f_q}], \tag{17}$$

where $Z \in \mathbb{R}^{m \times pq}$. The feature layer output $Z$ serves as the input to the enhancement layer.

**Enhancement Layer:** The enhancement layer enriches the feature representation by applying additional random transformations and a non-linear function to the concatenated feature nodes $Z$. Each window $(e_j)$ of enhancement nodes (there are $s$ windows, each with $r$ nodes) is computed as:

$$E_{e_j} = \psi(ZW_{e_j} + B_{e_j}), \tag{18}$$

where $Z \in \mathbb{R}^{m \times pq}$ is the output from the feature layer, $W_{e_j} \in \mathbb{R}^{pq \times r}$ is a random weight matrix, $B_{e_j} \in \mathbb{R}^{m \times r}$ is a bias matrix, $\psi(\cdot)$ is a non-linear function, and $E_{e_j} \in \mathbb{R}^{m \times r}$ is the output of the $j^{th}$ enhancement node window. The outputs of all enhancement windows are concatenated to form the overall enhancement layer output:

$$E = [E_{e_1}, E_{e_2}, \ldots, E_{e_s}], \tag{19}$$

where $E \in \mathbb{R}^{m \times rs}$.

**Output Layer:** The final representation matrix $A$, which combines the outputs of the feature and enhancement layers, is given by:

$$A = [Z \mid E], \tag{20}$$

where $A \in \mathbb{R}^{m \times (pq+rs)}$.

The output weights $\Theta \in \mathbb{R}^{(pq+rs) \times n_{\text{class}}}$ are then learned using a closed-form solution:

$$\Theta = (A^\top A + \lambda I)^{-1} A^\top Y, \tag{21}$$

where $\lambda$ is a regularization parameter.

**Incremental Learning:** One of BLS's unique features is its ability to incrementally update the network by adding new feature mapping nodes or enhancement nodes without retraining the entire model. This makes BLS computationally efficient and adaptable to streaming or evolving data.

### A.3 Formulation and Architecture of deep RVFL Shi et al. (2021)

The deep random vector functional link network (dRVFL) extends the standard RVFL architecture by incorporating multiple hidden layers, thereby enhancing its representational capacity while preserving the hallmark efficiency of closed-form training. Unlike conventional deep neural networks, dRVFL stacks several RVFL-like layers where only the final layer's output weights are trained via least squares, and all intermediate transformations are computed using fixed, randomly initialized weights. The architecture can be described as follows:

**Layer-wise Random Transformations:** Let the input matrix be $X \in \mathbb{R}^{m \times d}$. The $l^{\text{th}}$ hidden layer transformation is defined as:

$$H^{(l)} = \phi(H^{(l-1)} W^{(l)} + B^{(l)}), \quad l = 1, 2, \ldots, L, \tag{22}$$

where $H^{(0)} = X$, $W^{(l)} \in \mathbb{R}^{h_{l-1} \times h_l}$ is the randomly initialized weight matrix for layer $l$, $B^{(l)} \in \mathbb{R}^{m \times h_l}$ is the corresponding bias matrix, and $\phi(\cdot)$ is a nonlinear activation function applied elementwise.

**Augmented Feature Representation:** The final feature representation $A$ is constructed by concatenating the original input and the outputs of all hidden layers:

$$A = [X \mid H^{(1)} \mid H^{(2)} \mid \ldots \mid H^{(L)}] \in \mathbb{R}^{m \times a} \tag{23}$$

where $a = d + \sum_{l=1}^{L} h_l$ is the total feature dimensionality. This dense aggregation of features enables dRVFL to capture increasingly abstract representations at deeper layers while retaining the original input.

**Output Layer:** The output weights $\Theta \in \mathbb{R}^{a \times n_{\text{class}}}$ are obtained using the same closed-form solution as in RVFL:

$$\Theta = (A^\top A + \lambda I)^{-1} A^\top Y, \tag{24}$$

where $\lambda \geq 0$ is the regularization parameter.

## B Theoretical Analysis

**Lemma 1** (Spectral Conditioning via **StaR**). *Let $W \in \mathbb{R}^{d \times h}$ be a randomly initialized weight matrix with singular value decomposition $W = U\Sigma V^\top$, and let $W' = U\Sigma'V^\top$ be the spectrally regulated matrix obtained via the **StaR** procedure, where each singular value $\sigma_i' \in [\sigma_{low}, \sigma_{high}]$ for fixed bounds $0 < \sigma_{low} < \sigma_{high}$. Then, the regulated matrix $W'$ satisfies:*

1. *Spectral norm:* $\|W'\|_2 \leq \sigma_{high}$,

2. *Minimal singular value:* $\sigma_{\min}(W') \geq \sigma_{low}$,

3. *Condition number:* $\kappa(W') = \frac{\|W'\|_2}{\sigma_{\min}(W')} \leq \frac{\sigma_{high}}{\sigma_{low}}$.

*Proof.* By construction, the **StaR** framework rescales the singular values $\{\sigma_i\}$ of $W$ to lie within the interval $[\sigma_{\text{low}}, \sigma_{\text{high}}]$. Therefore, the resulting matrix $\Sigma' = \text{diag}(\sigma_1', \dots, \sigma_r')$ satisfies:

$$\sigma_{\min}(W') = \min_i \sigma_i' \geq \sigma_{\text{low}}, \quad \sigma_{\max}(W') = \max_i \sigma_i' \leq \sigma_{\text{high}}.$$

Hence, the operator norm $\|W'\|_2 = \sigma_{\max}(W') \leq \sigma_{\text{high}}$, and the condition number $\kappa(W') = \frac{\sigma_{\max}}{\sigma_{\min}} \leq \frac{\sigma_{\text{high}}}{\sigma_{\text{low}}}$, which completes the proof. $\square$

**Lemma 2** (Lipschitz Stability of the **StaR**-Transformed Mapping). *Let $\phi : \mathbb{R} \to \mathbb{R}$ be an $L_\phi$-Lipschitz activation function applied elementwise, and let $W' \in \mathbb{R}^{d \times h}$ be a **StaR**-regularized weight matrix satisfying $\|W'\|_2 \leq \sigma_{high}$. Then the hidden transformation $X \mapsto \phi(XW' + B)$ is Lipschitz continuous with constant $L_\phi \cdot \sigma_{high}$, i.e.,*

$$\|\phi(X_1 W' + B) - \phi(X_2 W' + B)\|_F \leq L_\phi \cdot \sigma_{high} \cdot \|X_1 - X_2\|_F$$

*for all $X_1, X_2 \in \mathbb{R}^{m \times d}$ and fixed bias matrix $B \in \mathbb{R}^{m \times h}$.*

*Proof.* By the Lipschitz continuity of $\phi$, applied elementwise,

$$\|\phi(X_1 W' + B) - \phi(X_2 W' + B)\|_F \leq L_\phi \cdot \|(X_1 - X_2)W'\|_F.$$

Using the submultiplicativity of the Frobenius norm:

$$\|(X_1 - X_2)W'\|_F \leq \|X_1 - X_2\|_F \cdot \|W'\|_2.$$

Since **StaR** ensures $\|W'\|_2 \leq \sigma_{\text{high}}$, we obtain:

$$\|\phi(X_1 W' + B) - \phi(X_2 W' + B)\|_F \leq L_\phi \cdot \sigma_{\text{high}} \cdot \|X_1 - X_2\|_F,$$

which concludes the proof. $\square$

**Lemma 3** (Supervised Warm-up Improves Output Prediction). *Let $H^{(0)} = \phi(XW^{(0)} + B^{(0)}) \in \mathbb{R}^{m \times h}$ be the initial hidden representation obtained from randomly initialized weights and biases, and let $\Theta^{(0)}$ be the corresponding closed-form output weights:*

$$\Theta^{(0)} = \arg\min_\Theta \|H^{(0)}\Theta - Y\|_F^2 + \lambda\|\Theta\|_F^2.$$

*Suppose that during the warm-up phase of **FLAiR**, the weights $W$ and biases $B$ are updated for $T$ steps using the loss*

$$\mathcal{L}^{(t)} = \|H^{(t)}\Theta^{(t)} - Y\|_F^2,$$

*where $H^{(t)} = \phi(XW^{(t)} + B^{(t)})$ and $\Theta^{(t)}$ is the closed-form solution at step $t$. Then, under the assumption that $\phi$ is differentiable and updates are performed using gradient descent with sufficiently small step size, the final prediction error after warm-up satisfies:*

$$\|H^{(T)}\Theta^{(T)} - Y\|_F^2 \leq \|H^{(0)}\Theta^{(0)} - Y\|_F^2.$$

*Proof.* At each warm-up step $t$, the output weights $\Theta^{(t)}$ are computed as the minimizer of the regularized least squares problem:

$$\Theta^{(t)} = (H^{(t)\top}H^{(t)} + \lambda I_h)^{-1}H^{(t)\top}Y.$$

The loss function minimized via backpropagation is:

$$\mathcal{L}^{(t)} = \|H^{(t)}\Theta^{(t)} - Y\|_F^2.$$

The gradient descent step on $W^{(t)}$, $B^{(t)}$ aims to reduce this loss at every iteration. Under standard smoothness assumptions on $\phi$ (e.g., Lipschitz continuity of its gradient), and for small enough learning rate, gradient descent ensures monotonic decrease in the training loss:

$$\mathcal{L}^{(t+1)} \leq \mathcal{L}^{(t)}, \quad \forall t.$$

Thus, by induction over $T$ steps, we obtain:

$$\mathcal{L}^{(T)} \leq \mathcal{L}^{(0)}.$$

Since $\mathcal{L}^{(t)} = \|H^{(t)}\Theta^{(t)} - Y\|_F^2$, this yields the desired result:

$$\|H^{(T)}\Theta^{(T)} - Y\|_F^2 \leq \|H^{(0)}\Theta^{(0)} - Y\|_F^2.$$

$\square$

## C  EXPERIMENTAL SETUP AND HYPERPARAMETER SETTING

**Experimental Setup:** All experiments for the baseline RVFL, ELM, and BLS models, along with their **StaR**-enhanced counterparts, are implemented using MATLAB R2023a and executed on a Windows 10 PC equipped with an Intel(R) Core(TM) i7-6700 CPU @ 3.40GHz (4 cores, 8 logical processors) and 16 GB RAM. The dRVFL model and its **StaR**-enhanced version, as well as all **FLAiR**-enhanced models and their corresponding baselines, are implemented in Python 3.12.7 within a Conda environment (version 24.11.3) and executed using Visual Studio Code on the same hardware configuration. Each dataset is preprocessed by normalizing the input features to have zero mean and unit variance. The details of all binary and multiclass datasets used in the experiments are provided in Table 3. A 5-fold cross-validation procedure is employed to ensure reliable and unbiased evaluation. In each fold, the dataset is split into $80\%$ training data and $20\%$ testing data. For every combination of hyperparameters, the model is trained on the training data and evaluated on the testing data across all 5 folds. The testing accuracy is recorded for each fold. The final testing accuracy for each dataset is computed as the mean testing accuracy across the five folds, providing a robust estimate of the model's performance. Additionally, the standard deviation of testing accuracy across the folds is recorded to quantify the stability of the model.

**Hyperparameter Setting:** Hyperparameter tuning is performed using a grid search strategy to identify the optimal settings for each model. For each model, the regularization parameter ($\lambda$) is selected from $\{10^i \mid i = -5, -4, \ldots, 5\}$. For RVFL and ELM, the number of hidden nodes ($h$) varies from $[3 : 20 : 203]$, and six activation functions (Sigmoid (1), Sine (2), Tribas (3), Radbas (4), Tansig (5), and ReLU (6)) are evaluated. For BLS, the number of feature windows ($q$), the number of feature nodes in each window ($p$), and the number of enhancement nodes ($r$) are set as per Sajid et al. (2024a), with Tansig as the activation function. For dRVFL, we adopt the same hyperparameter settings as provided in Shi et al. (2021) and evaluate three activation functions: Sigmoid (1), ReLU (2), and SELU (3). For the **StaR**-enhanced counterparts, the spectral bounds are selected by varying the upper threshold $\sigma_{\text{high}}$ in the range $\{0.5, 0.75, 1.0, 1.25, 1.5, 1.75, 2.0\}$, with the corresponding lower bound set as $\sigma_{\text{low}} = 0.1 \cdot \sigma_{\text{high}}$. For the **FLAiR**-enhanced counterparts, the number of warm-up epochs is chosen from the range $\{2, 3, \ldots, 10\}$, enabling progressive refinement of the input-to-hidden weights via lightweight adaptation. All **FLAiR** models are trained using the standard Adam optimizer Kingma & Ba (2017) for backpropagation during the warm-up phase.

## D  DATASET-WISE RESULTS ON BINARY AND MULTICLASS DATASETS

The experimental results are comprehensively summarized in Tables 4 - 9. Tables 4 and 5 compare the baseline RVFL, ELM, and BLS models with their **StaR**-enhanced counterparts on binary and multiclass classification tasks, respectively. Tables 6 and 7 present similar comparisons for the **FLAiR**-enhanced versions. Furthermore, Tables 8 and 9 provide the performance of dRVFL and its enhancements using both the **StaR** and **FLAiR** frameworks on binary and multiclass datasets, respectively.

## E  STATISTICAL ANALYSIS

To statistically validate the observed improvements, we conduct multiple statistical tests, including the Friedman test, the Nemenyi post-hoc test, and the win–tie–loss analysis Demšar (2006).

Table 3: Summary of all datasets used in the experiments. The table lists the dataset name, number of samples, number of features, and number of classes.

| Dataset | Number of Samples | Number of Features | Number of Classes | Dataset | Number of Samples | Number of Features | Number of Classes |
|---|---|---|---|---|---|---|---|
| **Binary datasets** | | | | | | | |
| acute_inflammation | 120 | 6 | 2 | monk2 | 601 | 7 | 2 |
| acute_nephritis | 120 | 6 | 2 | monk3 | 554 | 6 | 2 |
| balloons | 16 | 4 | 2 | new-thyroid1 | 215 | 5 | 2 |
| fertility | 100 | 9 | 2 | pima | 768 | 8 | 2 |
| parkinsons | 195 | 22 | 2 | ripley | 1250 | 2 | 2 |
| pittsburg_bridges_T_OR_D | 102 | 7 | 2 | segment0 | 2308 | 19 | 2 |
| bank | 4521 | 16 | 2 | shuttle-6_vs_2-3 | 230 | 9 | 2 |
| blood | 748 | 4 | 2 | shuttle-c0-vs-c4 | 1829 | 9 | 2 |
| breast_cancer | 286 | 9 | 2 | sonar | 208 | 60 | 2 |
| breast_cancer_wisc | 699 | 9 | 2 | votes | 435 | 16 | 2 |
| breast_cancer_wisc_diag | 569 | 30 | 2 | vowel | 988 | 10 | 2 |
| breast_cancer_wisc_prog | 198 | 33 | 2 | wpbc | 194 | 33 | 2 |
| chess_krvkp | 3196 | 36 | 2 | yeast-0-2-5-6_vs_3-7-8-9 | 1004 | 8 | 2 |
| congressional_voting | 435 | 16 | 2 | yeast-0-2-5-7-9_vs_3-6-8 | 1004 | 8 | 2 |
| conn_bench_sonar_mines_rocks | 208 | 60 | 2 | yeast-0-5-6-7-9_vs_4 | 528 | 8 | 2 |
| credit_approval | 690 | 15 | 2 | yeast-2_vs_4 | 514 | 8 | 2 |
| cylinder_bands | 512 | 35 | 2 | yeast1vs7 | 459 | 8 | 2 |
| echocardiogram | 131 | 10 | 2 | yeast2vs8 | 483 | 8 | 2 |
| haberman_survival | 306 | 3 | 2 | yeast3 | 1484 | 8 | 2 |
| hepatitis | 155 | 19 | 2 | **Dataset** | **Number of Samples** | **Number of Features** | **Number of Classes** |
| horse_colic | 368 | 25 | 2 | **Multiclass Dataset** | | | |
| ilpd_indian_liver | 583 | 9 | 2 | abalone | 4177 | 8 | 3 |
| ionosphere | 351 | 33 | 2 | annealing | 898 | 31 | 5 |
| monks_1 | 556 | 6 | 2 | arrhythmia | 452 | 262 | 13 |
| monks_3 | 554 | 6 | 2 | balance_scale | 625 | 4 | 3 |
| oocytes_merluccius_nucleus_4d | 1022 | 41 | 2 | cardiotocography_10clases | 2126 | 21 | 10 |
| oocytes_trisopterus_nucleus_2f | 912 | 25 | 2 | cardiotocography_3clases | 2126 | 21 | 3 |
| pima | 768 | 8 | 2 | conn_bench_vowel_deterding | 990 | 11 | 11 |
| planning | 182 | 12 | 2 | contrac | 1473 | 9 | 3 |
| spect | 265 | 22 | 2 | dermatology | 366 | 34 | 6 |
| spectf | 267 | 44 | 2 | ecoli | 336 | 7 | 8 |
| statlog_australian_credit | 690 | 14 | 2 | energy_y1 | 768 | 8 | 3 |
| statlog_german_credit | 1000 | 24 | 2 | energy_y2 | 768 | 8 | 3 |
| statlog_heart | 270 | 13 | 2 | flags | 194 | 28 | 8 |
| tic_tac_toe | 958 | 9 | 2 | glass | 214 | 9 | 6 |
| titanic | 2201 | 3 | 2 | heart_cleveland | 303 | 13 | 5 |
| adult | 48842 | 14 | 2 | heart_switzerland | 123 | 12 | 5 |
| connect_4 | 67557 | 42 | 2 | image_segmentation | 2310 | 18 | 7 |
| hill_valley | 1212 | 100 | 2 | iris | 150 | 4 | 3 |
| magic | 19020 | 10 | 2 | led_display | 1000 | 7 | 10 |
| miniboone | 130064 | 50 | 2 | lenses | 24 | 4 | 3 |
| mushroom | 8124 | 21 | 2 | letter | 20000 | 16 | 26 |
| musk_2 | 6598 | 166 | 2 | low_res_spect | 531 | 100 | 9 |
| ozone | 2536 | 72 | 2 | lymphography | 148 | 18 | 4 |
| spambase | 4601 | 57 | 2 | molec_biol_splice | 3190 | 60 | 3 |
| twonorm | 7400 | 20 | 2 | nursery | 12960 | 8 | 5 |
| abalone9-18 | 731 | 7 | 2 | oocytes_merluccius_states_2f | 1022 | 25 | 3 |
| aus | 690 | 14 | 2 | oocytes_trisopterus_states_5b | 912 | 32 | 3 |
| brwisconsin | 683 | 9 | 2 | optical | 5620 | 62 | 10 |
| bupa or liver-disorders | 345 | 6 | 2 | page_blocks | 5473 | 10 | 5 |
| checkerboard_Data | 690 | 14 | 2 | pendigits | 10992 | 16 | 10 |
| cleve | 297 | 13 | 2 | pittsburg_bridges_MATERIAL | 106 | 7 | 3 |
| crossplane150 | 150 | 2 | 2 | pittsburg_bridges_REL_L | 103 | 7 | 3 |
| ecoli-0-1-4-6_vs_5 | 280 | 6 | 2 | pittsburg_bridges_SPAN | 92 | 7 | 3 |
| ecoli-0-1-4-7_vs_2-3-5-6 | 336 | 7 | 2 | pittsburg_bridges_TYPE | 105 | 7 | 6 |
| ecoli-0-1_vs_2-3-5 | 244 | 7 | 2 | post_operative | 90 | 8 | 3 |
| ecoli-0-1_vs_5 | 240 | 6 | 2 | seeds | 210 | 7 | 3 |
| ecoli-0-2-3-4_vs_5 | 202 | 7 | 2 | semeion | 1593 | 256 | 10 |
| ecoli-0-2-6-7_vs_3-5 | 224 | 7 | 2 | soybean | 683 | 35 | 18 |
| ecoli-0-3-4-6_vs_5 | 205 | 7 | 2 | statlog_image | 2310 | 18 | 7 |
| ecoli-0-3-4-7_vs_5-6 | 257 | 7 | 2 | statlog_landsat | 6435 | 36 | 6 |
| ecoli-0-4-6_vs_5 | 203 | 6 | 2 | statlog_shuttle | 58000 | 9 | 7 |
| ecoli-0-6-7_vs_5 | 220 | 6 | 2 | statlog_vehicle | 846 | 18 | 4 |
| ecoli0137vs26 | 311 | 7 | 2 | synthetic_control | 600 | 60 | 6 |
| ecoli01vs5 | 240 | 7 | 2 | thyroid | 7200 | 21 | 3 |
| ecoli3 | 336 | 7 | 2 | vertebral_column_3clases | 310 | 6 | 3 |
| ecoli4 | 336 | 7 | 2 | wall_following | 5456 | 24 | 4 |
| glass2 | 214 | 9 | 2 | waveform | 5000 | 21 | 3 |
| haber | 306 | 3 | 2 | waveform_noise | 5000 | 40 | 3 |
| haberman | 306 | 3 | 2 | wine | 178 | 13 | 3 |
| heart-stat | 270 | 13 | 2 | wine_quality_red | 1599 | 11 | 6 |
| iono | 351 | 33 | 2 | wine_quality_white | 4898 | 11 | 7 |
| led7digit-0-2-4-5-6-7-8-9_vs_1 | 443 | 7 | 2 | yeast | 1484 | 8 | 10 |
| monk1 | 556 | 6 | 2 | zoo | 101 | 16 | 7 |

The Friedman test results, presented in Tables 10 and 11, confirm that the StaR- and FLAiR-enhanced models consistently achieve statistically significant improvements over their respective baselines. According to the null hypothesis, all models are considered equivalent, implying that their average ranks do not significantly differ. The Friedman test statistic, which adheres to a chi-squared ($\chi_F^2$) distribution with $l-1$ degrees of freedom (where $l$ represents the total number of models), is computed as $\chi_F^2 = \frac{12\mathcal{D}}{l(l+1)} \left[ \sum_k R_k^2 - \frac{l(l+1)^2}{4} \right]$, where $\mathcal{D}$ represents the number of datasets, and $R_k$ denotes the mean rank of the $k^{th}$ model. Since the Friedman statistic is often overly conservative, an improved statistic proposed by Iman & Davenport (1980) provides a more accurate measure: $F_F = \frac{(\mathcal{D}-1)\chi_F^2}{\mathcal{D}(l-1)-\chi_F^2}$, which follows the $F$-distribution with degrees of freedom $(l-1)$ and $(l-1)(\mathcal{D}-1)$. At a 5% significance level, the critical values of $F((l-1),(l-1)(\mathcal{D}-1))$ are obtained from the $F$-distribution table. For RVFL, ELM, and BLS, the critical values are 3.94455 for binary datasets and 4.0266 for multi-class datasets. For dRVFL, the corresponding critical values are 4.1213 and 4.13, respectively. Since $F_F > F((l-1),(l-1)(\mathcal{D}-1))$ for all cases, the null hypothesis is rejected, confirming that the performance differences among the models are statistically significant.

Table 4: Comparison of RVFL, ELM, and BLS with their **StaR**-enhanced counterparts on binary classification tasks.

| Dataset | RVFL | | StaR-RVFL | | ELM | | StaR-ELM | | BLS | | StaR-BLS | |
|---|---|---|---|---|---|---|---|---|---|---|---|---|
| | Acc. | Std. | Acc. | Std. | Acc. | Std. | Acc. | Std. | Acc. | Std. | Acc. | Std. |
| acute_nephritis | 100 | 0 | 100 | 0 | 100 | 0 | 100 | 0 | 100 | 0 | 100 | 0 |
| balloons | 75 | 14.4338 | 86.6667 | 19.8142 | 73.3333 | 27.8887 | 86.6667 | 18.2574 | 86.6667 | 18.2574 | 86.6667 | 11.2574 |
| fertility | 91 | 8.2158 | 91 | 8.2158 | 91 | 8.2158 | 91 | 8.2158 | 90 | 10 | 91 | 8.2158 |
| parkinsons | 83.0769 | 12.8972 | 84.1026 | 10.7875 | 82.0513 | 17.6719 | 83.5897 | 17.4472 | 81.5385 | 16.3581 | 84.6154 | 10.4154 |
| pittsburg_bridges_T_OR_D | 88.2381 | 8.9271 | 91.1905 | 3.9912 | 87.2381 | 10.261 | 90.1905 | 4.8894 | 91.1905 | 6.3976 | 90.1905 | 6.1293 |
| bank | 89.6926 | 0.4758 | 89.936 | 0.5773 | 89.5377 | 0.3955 | 89.9139 | 0.3463 | 89.7809 | 0.4067 | 90.0905 | 0.6165 |
| blood | 76.5056 | 13.5754 | 77.7047 | 12.4533 | 76.7723 | 13.3533 | 77.4371 | 12.5572 | 77.9723 | 11.6847 | 78.1065 | 10.793 |
| breast_cancer | 70.1754 | 44.6249 | 70.1754 | 32.4134 | 70.1754 | 44.6249 | 70.1754 | 32.2805 | 70.2057 | 26.0229 | 73.3333 | 22.3932 |
| breast_cancer_wisc | 88.4183 | 6.7943 | 88.9866 | 4.297 | 88.4152 | 3.6106 | 89.1305 | 3.076 | 88.1336 | 5.2028 | 88.4193 | 4.9471 |
| breast_cancer_wisc_diag | 94.549 | 2.5218 | 95.6078 | 1.7506 | 94.0273 | 1.8937 | 95.2538 | 2.2902 | 94.3798 | 3.2536 | 94.5536 | 2.7283 |
| breast_cancer_wisc_prog | 81.3846 | 8.341 | 82.359 | 5.2159 | 80.8974 | 9.027 | 82.8718 | 4.3114 | 83.3846 | 8.4335 | 81.3846 | 7.0594 |
| chess_krvkp | 81.6346 | 7.0558 | 83.1976 | 4.8692 | 80.4456 | 6.8631 | 82.3852 | 5.4534 | 85.2316 | 4.0604 | 85.0434 | 3.1451 |
| congressional_voting | 63.4483 | 4.9036 | 63.908 | 2.5183 | 62.9885 | 4.0962 | 64.1379 | 3.4864 | 63.6782 | 2.885 | 64.3678 | 3.8084 |
| conn_bench_sonar_mines_rocks | 60.1161 | 12.8152 | 64.4019 | 2.9293 | 62.5436 | 8.566 | 63.4262 | 6.2516 | 68.3391 | 9.7684 | 71.1614 | 6.0878 |
| credit_approval | 85.3623 | 12.1838 | 85.6522 | 10.355 | 85.0725 | 11.4415 | 85.7971 | 8.8542 | 86.6667 | 10.3702 | 87.1014 | 9.125 |
| cylinder_bands | 67.5919 | 5.0677 | 70.3122 | 5.2431 | 67.7803 | 3.7032 | 69.9277 | 2.536 | 69.5298 | 3.8847 | 69.9219 | 3.4197 |
| echocardiogram | 84.6724 | 7.2724 | 85.4416 | 6.3987 | 84.7009 | 5.5176 | 86.2108 | 7.0505 | 85.4416 | 6.3987 | 85.4416 | 6.4657 |
| haberman_survival | 73.4902 | 8.4751 | 73.4902 | 8.4751 | 73.4902 | 8.4751 | 73.8181 | 8.4202 | 70.6029 | 6.662 | 73.4902 | 8.4751 |
| hepatitis | 84.5161 | 12.7817 | 85.1613 | 8.7156 | 83.2258 | 9.2373 | 86.4516 | 8.0467 | 85.1613 | 10.6011 | 85.8065 | 9.8374 |
| horse_colic | 86.1385 | 3.7694 | 86.4198 | 3.281 | 86.4235 | 4.9395 | 86.6753 | 2.6833 | 86.6938 | 3.1932 | 87.2381 | 3.7511 |
| ilpd_indian_liver | 71.7138 | 6.9558 | 72.903 | 4.2655 | 71.8744 | 4.0378 | 71.8671 | 5.5082 | 72.0395 | 4.6401 | 72.9016 | 5.5345 |
| ionosphere | 90.33 | 5.7816 | 90.6278 | 6.9368 | 89.7626 | 6.0272 | 90.6117 | 5.7258 | 88.3541 | 9.5467 | 89.2032 | 8.634 |
| monks_1 | 81.6297 | 13.7116 | 83.4363 | 8.6492 | 83.7902 | 9.1321 | 82.7107 | 9.1082 | 74.6268 | 5.3631 | 76.6007 | 4.4493 |
| monks_3 | 91.1581 | 4.0925 | 91.8804 | 3.8683 | 90.9812 | 4.3104 | 92.0622 | 5.3771 | 87.5414 | 4.6258 | 87.1957 | 5.9057 |
| oocytes_merluccius_nucleus_4d | 82.5868 | 3.0751 | 83.66 | 2.9505 | 82.0937 | 3.1457 | 83.3649 | 1.9663 | 82.4859 | 2.1099 | 83.0722 | 1.8323 |
| oocytes_trisopterus_nucleus_2f | 79.2686 | 4.1506 | 80.9127 | 5.4171 | 78.5096 | 3.8063 | 80.5855 | 3.3711 | 78.9443 | 2.073 | 80.0408 | 2.6117 |
| pima | 72.532 | 4.4896 | 75.0055 | 3.9788 | 73.0473 | 1.4865 | 74.4843 | 2.5672 | 72.2655 | 3.8872 | 72.5236 | 3.2635 |
| planning | 71.3814 | 8.8534 | 71.9369 | 8.2169 | 71.9369 | 9.2169 | 71.9369 | 7.1382 | 71.3814 | 8.8534 | 73.5886 | 7.1156 |
| spect | 67.9245 | 7.893 | 68.6792 | 3.1572 | 67.9245 | 6.3984 | 68.3019 | 3.3752 | 69.0566 | 4.1338 | 69.8113 | 4.2243 |
| spectf | 79.3431 | 20.8936 | 79.3431 | 20.8936 | 79.7205 | 20.7374 | 79.3431 | 16.8936 | 79.3431 | 20.8936 | 79.7205 | 20.7374 |
| statlog_australian_credit | 68.2609 | 1.0748 | 68.5507 | 0.8262 | 68.5507 | 2.9256 | 68.5507 | 1.3939 | 68.1159 | 2.8065 | 68.9855 | 1.571 |
| statlog_german_credit | 77 | 4.3589 | 77.4 | 1.917 | 76.8 | 4.0404 | 77.4 | 3.3801 | 77.1 | 2.2749 | 77 | 1.7678 |
| statlog_heart | 81.1111 | 3.5621 | 81.4815 | 2.343 | 80.3704 | 5.4935 | 81.4815 | 5.0715 | 82.963 | 3.3127 | 83.3333 | 2.6189 |
| tic_tac_toe | 88.3017 | 9.0423 | 93.101 | 8.862 | 87.987 | 10.7038 | 91.7479 | 7.8709 | 98.1195 | 1.9779 | 98.2226 | 1.9209 |
| titanic | 77.3259 | 16.0188 | 77.9168 | 15.5828 | 77.6901 | 15.4985 | 79.0532 | 15.0381 | 77.9168 | 15.5828 | 78.4623 | 10.2711 |
| adult | 84.282 | 0.1975 | 84.6505 | 0.1594 | 84.2144 | 0.2149 | 84.63 | 0.1945 | 84.327 | 0.279 | 84.3495 | 0.2315 |
| connect_4 | 75.5569 | 3.7062 | 75.7464 | 3.5404 | 75.4592 | 3.78 | 75.7153 | 4.2827 | 75.511 | 3.8167 | 75.6102 | 3.7243 |
| hill_valley | 81.4318 | 4.3094 | 82.1763 | 4.5884 | 77.8904 | 5.3723 | 81.8485 | 4.9289 | 82.345 | 5.2034 | 82.9242 | 5.0465 |
| magic | 78.612 | 15.7624 | 79.4217 | 15.3015 | 78.3596 | 16.2135 | 79.4059 | 15.6682 | 76.6141 | 16.3535 | 76.9664 | 12.1802 |
| miniboone | 82.8393 | 18.4002 | 84.1671 | 17.2489 | 81.5984 | 19.1533 | 83.8526 | 17.3439 | 84.4339 | 16.2213 | 84.6453 | 15.5055 |
| mushroom | 97.3779 | 2.5765 | 99.0151 | 1.2892 | 96.3073 | 3.9179 | 98.855 | 1.6016 | 98.7442 | 1.7465 | 98.8672 | 1.8249 |
| musk_2 | 84.5909 | 34.4558 | 84.5909 | 24.4228 | 84.5909 | 34.4558 | 85.6212 | 22.1519 | 84.5909 | 34.4558 | 85.317 | 26.4714 |
| ozone | 97.1217 | 2.2639 | 97.1217 | 2.1594 | 97.1611 | 2.2511 | 97.1611 | 2.2511 | 97.1611 | 2.2511 | 97.24 | 2.0916 |
| spambase | 88.481 | 4.7026 | 89.2203 | 4.1411 | 88.1775 | 5.2705 | 88.612 | 5.1656 | 89.1985 | 4.1898 | 89.6106 | 2.0493 |
| twonorm | 51.3514 | 0.9053 | 51.4459 | 1.3519 | 51.6216 | 1.1615 | 51.5811 | 1.0593 | 52.1892 | 0.7946 | 52.2432 | 0.584 |
| abalone9-18 | 95.9035 | 3.9418 | 96.0404 | 3.8631 | 95.9025 | 3.5716 | 96.0404 | 3.4631 | 96.1765 | 3.6227 | 96.3126 | 3.3924 |
| aus | 86.3768 | 3.9157 | 86.9565 | 2.5102 | 86.3768 | 3.5277 | 86.8116 | 4.3899 | 85.6522 | 2.5824 | 86.087 | 2.6736 |
| brwisconsin | 90.629 | 2.6065 | 90.9221 | 3.6967 | 90.4841 | 6.0049 | 91.213 | 4.6945 | 91.6595 | 4.8009 | 91.9483 | 4.6245 |
| bupa or liver-disorders | 73.3333 | 4.5369 | 74.2029 | 3.3401 | 73.913 | 2.7113 | 73.6232 | 1.6286 | 74.7826 | 4.1753 | 74.7826 | 4.2992 |
| checkerboard_Data | 85.942 | 4.3053 | 86.9565 | 2.7593 | 86.087 | 3.2568 | 86.8116 | 2.4786 | 85.5072 | 2.562 | 85.6522 | 2.3756 |
| cleve | 81.1469 | 2.4849 | 81.8023 | 1.7427 | 80.4576 | 4.2873 | 82.1469 | 2.6213 | 82.8192 | 2.8433 | 83.4859 | 3.9214 |
| crossplane150 | 62 | 12.1564 | 62 | 12.1564 | 62 | 12.1564 | 62 | 10.0286 | 62 | 12.1564 | 62 | 12.1564 |
| ecoli-0-1-4-6_vs_5 | 98.5714 | 1.494 | 98.9286 | 1.3972 | 98.9286 | 1.5972 | 98.9286 | 1.5972 | 98.9286 | 1.5972 | 99.2857 | 1.5972 |
| ecoli-0-1-4-7_vs_2-3-5-6 | 97.9104 | 1.335 | 97.9104 | 1.335 | 97.3178 | 0.6773 | 97.9104 | 1.3435 | 97.9192 | 1.3277 | 98.5075 | 1.0554 |
| ecoli-0-1_vs_2-3-5 | 97.9592 | 3.5348 | 98.3673 | 2.6609 | 97.9592 | 3.5348 | 98.3673 | 2.6609 | 98.7755 | 1.8254 | 98.7755 | 1.8254 |
| ecoli-0-1_vs_5 | 98.75 | 1.1411 | 98.75 | 1.1411 | 97.9167 | 2.0833 | 98.75 | 1.8634 | 99.1667 | 1.1411 | 98.75 | 1.1411 |
| ecoli-0-2-3-4_vs_5 | 98.5122 | 2.2294 | 99.5 | 1.118 | 98.5 | 2.2361 | 99 | 2.2361 | 99 | 2.2361 | 99.0122 | 1.3528 |
| ecoli-0-2-6-7_vs_3-5 | 96.8889 | 5.7948 | 97.3333 | 5.5628 | 97.3232 | 4.8166 | 97.3333 | 4.6177 | 98.2222 | 3.9752 | 98.2222 | 3.9752 |
| ecoli-0-3-4-6_vs_5 | 98.0488 | 2.0406 | 98.5366 | 2.1815 | 98.0488 | 2.6718 | 98.5366 | 2.1815 | 98.5366 | 2.1815 | 99.0244 | 1.3359 |
| ecoli-0-3-4-7_vs_5-6 | 98.8311 | 1.0672 | 98.8311 | 1.0672 | 98.8311 | 1.0672 | 98.8311 | 1.0672 | 98.8311 | 1.0672 | 98.8311 | 1.0672 |
| ecoli-0-4-6_vs_5 | 98.5122 | 1.3584 | 98.5244 | 2.1885 | 98.5244 | 1.3473 | 98.5244 | 1.1885 | 99 | 1.3693 | 99.0122 | 1.3528 |
| ecoli-0-6-7_vs_5 | 98.1818 | 1.9015 | 98.1818 | 1.0164 | 97.7273 | 1.6071 | 98.1818 | 1.0164 | 98.1818 | 1.9015 | 98.6364 | 1.2448 |
| ecoli0137vs26 | 96.4772 | 3.0561 | 97.1173 | 2.6162 | 96.7998 | 2.9663 | 97.1173 | 2.6162 | 96.7998 | 2.9663 | 97.1173 | 2.3545 |
| ecoli01vs5 | 99.5833 | 0.9317 | 99.5833 | 0.9317 | 100 | 0 | 99.5833 | 0.9317 | 100 | 0 | 100 | 0 |
| ecoli3 | 94.3371 | 4.4091 | 94.3459 | 4.0035 | 94.043 | 4.2275 | 94.3459 | 3.4018 | 94.3459 | 3.4018 | 94.9385 | 3.0983 |
| ecoli4 | 99.1089 | 0.8135 | 99.1089 | 0.8135 | 98.8147 | 1.2331 | 99.1089 | 0.8135 | 99.1089 | 0.8135 | 99.403 | 0.8175 |
| glass2 | 92.0487 | 2.1192 | 92.0487 | 2.1192 | 92.0487 | 2.1192 | 92.0487 | 2.1192 | 92.0487 | 2.1192 | 92.5138 | 2.0004 |
| haber | 74.146 | 8.0641 | 74.146 | 8.0641 | 74.146 | 8.0641 | 74.146 | 8.0641 | 74.146 | 8.0641 | 74.4685 | 6.6426 |
| haberman | 73.8181 | 8.0332 | 74.146 | 8.0641 | 74.146 | 8.0641 | 74.146 | 8.0641 | 74.146 | 8.0641 | 74.4685 | 6.6426 |
| heart-stat | 81.4815 | 3.7037 | 82.2222 | 4.4598 | 80 | 3.3127 | 82.963 | 3.5621 | 81.8519 | 3.0429 | 83.3333 | 2.268 |
| iono | 92.3219 | 5.9031 | 93.171 | 4.7581 | 94.0201 | 4.7751 | 93.7465 | 6.0777 | 90.0483 | 7.9003 | 92.0322 | 5.1838 |
| led7digit-0-2-4-5-6-7-8-9_vs_1 | 96.6216 | 1.58 | 96.8488 | 1.8393 | 96.6216 | 1.58 | 96.6241 | 2.0952 | 96.3968 | 1.8373 | 96.3968 | 1.8373 |
| monk1 | 52.1412 | 5.306 | 52.6786 | 6.0064 | 52.6786 | 6.0401 | 52.5064 | 3.836 | 53.2159 | 6.789 | 53.2207 | 5.2456 |
| monk2 | 68.2204 | 6.9017 | 67.5592 | 6.2258 | 67.7231 | 5.2935 | 67.719 | 6.5003 | 68.5551 | 7.5396 | 68.8829 | 6.8224 |
| monk3 | 52.5225 | 5.4391 | 52.1622 | 3.8802 | 52.3423 | 4.9998 | 52.7027 | 5.4054 | 53.2423 | 6.5926 | 53.0631 | 5.1596 |
| new-thyroid1 | 99.5349 | 1.04 | 98.6047 | 1.2738 | 99.5349 | 1.04 | 98.6047 | 1.2738 | 100 | 0 | 100 | 0 |
| pima | 71.0916 | 3.5385 | 71.089 | 4.2695 | 71.2206 | 2.5043 | 71.0941 | 3.5099 | 71.999 | 3.1541 | 72.2647 | 4.208 |
| ripley | 59.84 | 3.3657 | 59.84 | 3.3657 | 59.84 | 3.3657 | 59.84 | 3.3657 | 59.84 | 3.3657 | 59.84 | 3.3657 |
| segment0 | 99.6968 | 0.1184 | 99.6968 | 0.1184 | 99.6535 | 0.1934 | 99.6968 | 0.1184 | 99.7401 | 0.0967 | 99.7401 | 0.0967 |
| shuttle-6_vs_2-3 | 100 | 0 | 100 | 0 | 100 | 0 | 100 | 0 | 100 | 0 | 100 | 0 |
| shuttle-c0-vs-c4 | 100 | 0 | 100 | 0 | 100 | 0 | 100 | 0 | 100 | 0 | 100 | 0 |
| sonar | 85.5052 | 8.7141 | 85.5865 | 2.9101 | 83.1243 | 5.8574 | 85.0523 | 6.3716 | 86.0163 | 8.012 | 87.4913 | 2.0645 |
| votes | 97.0115 | 2.2406 | 97.0115 | 1.7682 | 96.7816 | 2.7442 | 97.4713 | 2.0562 | 97.0115 | 3.5983 | 97.0115 | 3.5983 |
| vowel | 96.4549 | 2.239 | 96.5559 | 2.0115 | 95.9509 | 1.132 | 96.8589 | 2.1659 | 96.5569 | 2.189 | 97.263 | 2.2884 |
| wpbc | 81.4305 | 3.8828 | 81.9568 | 3.0597 | 80.4184 | 4.2533 | 81.4575 | 5.5055 | 78.8664 | 11.0845 | 80.9042 | 8.1087 |
| yeast-0-2-5-6_vs_3-7-8-9 | 93.7214 | 2.2892 | 94.0214 | 2.0642 | 93.8204 | 2.4502 | 93.6219 | 2.3102 | 94.1199 | 2.2274 | 94.3199 | 1.897 |
| yeast-0-2-5-7-9_vs_3-6-8 | 97.4109 | 0.8157 | 97.4114 | 0.9549 | 97.4114 | 1.0767 | 97.4114 | 1.0767 | 97.4119 | 1.0767 | 97.5109 | 0.9288 |
| yeast-0-5-6-7-9_vs_4 | 93.5544 | 2.08 | 93.3711 | 1.3473 | 93.9389 | 1.0797 | 94.124 | 1.5831 | 93.9389 | 1.2758 | 93.9407 | 1.4239 |
| yeast-2_vs_4 | 96.6933 | 2.0119 | 96.6952 | 1.2141 | 97.0798 | 1.5412 | 97.0817 | 1.1891 | 97.0798 | 1.3798 | 97.0836 | 2.1688 |
| yeast1vs7 | 95.2102 | 1.6402 | 95.2102 | 1.6402 | 95.4276 | 1.409 | 95.2102 | 1.2284 | 95.645 | 1.324 | 95.8624 | 1.4118 |
| yeast2vs8 | 98.1379 | 1.8496 | 98.3462 | 1.5649 | 98.14 | 1.8448 | 98.3462 | 1.5649 | 98.5524 | 1.3856 | 98.5524 | 1.3856 |
| yeast3 | 94.8794 | 2.0656 | 95.1488 | 2.3868 | 94.8792 | 2.4891 | 95.0143 | 1.967 | 95.0817 | 2.1621 | 95.5531 | 2.4433 |

Table 5: Comparison of RVFL, ELM, and BLS with their **StaR**-enhanced counterparts on multiclass classification tasks.

| Dataset | RVFL Acc. | RVFL Std. | StaR-RVFL Acc. | StaR-RVFL Std. | ELM Acc. | ELM Std. | StaR-ELM Acc. | StaR-ELM Std. | BLS Acc. | BLS Std. | StaR-BLS Acc. | StaR-BLS Std. |
|---|---|---|---|---|---|---|---|---|---|---|---|---|
| abalone | 63.7295 | 1.8436 | 63.5858 | 2.724 | 63.586 | 1.4052 | 63.7533 | 1.5148 | 63.3458 | 2.2389 | 63.6094 | 2.1706 |
| annealing | 89.1899 | 5.2447 | 90.4146 | 5.2285 | 88.9677 | 3.9794 | 90.4146 | 3.66 | 90.0807 | 4.4372 | 90.1924 | 4.3932 |
| arrhythmia | 70.5812 | 3.4937 | 70.7985 | 3.2655 | 65.7118 | 6.1656 | 68.8181 | 5.8023 | 65.0379 | 2.9768 | 66.1392 | 4.1551 |
| balance_scale | 98.4 | 1.2649 | 97.92 | 1.2533 | 98.4 | 0.8 | 98.24 | 1.5388 | 98.08 | 1.3387 | 98.24 | 0.6693 |
| cardiotocography_10clases | 69.1463 | 4.3501 | 71.1208 | 4.2849 | 70.5569 | 4.4265 | 71.2154 | 4.3552 | 65.9468 | 4.6107 | 66.2289 | 3.4441 |
| cardiotocography_3clases | 85.421 | 7.4456 | 86.3153 | 7.2336 | 85.7049 | 8.1929 | 86.3624 | 7.9329 | 85.7514 | 7.8278 | 86.0325 | 7.048 |
| conn_bench_vowel_deterding | 95.6566 | 9.158 | 95.6566 | 8.0855 | 95.6566 | 7.8046 | 95.7576 | 7.1013 | 95.9596 | 8.755 | 96.2626 | 8.0777 |
| contrac | 40.6597 | 9.9532 | 41.2708 | 8.0619 | 40.7956 | 10.5615 | 41.3373 | 10.4955 | 44.4707 | 7.2873 | 49.0174 | 9.1506 |
| dermatology | 97.2714 | 1.6665 | 97.8156 | 1.5587 | 96.7308 | 2.8118 | 97.8119 | 2.0793 | 97.5379 | 2.2532 | 98.3562 | 1.7861 |
| ecoli | 61.2072 | 36.4683 | 61.5057 | 22.4895 | 60.619 | 36.0715 | 60.9131 | 24.4024 | 60.619 | 36.0715 | 61.5101 | 25.1872 |
| energy_y1 | 89.0578 | 5.2589 | 91.5355 | 4.4592 | 89.1868 | 2.8495 | 91.1442 | 2.277 | 87.8924 | 5.1619 | 89.1936 | 5.0621 |
| energy_y2 | 90.1002 | 4.1093 | 92.7052 | 3.2735 | 90.6205 | 5.5147 | 91.4057 | 4.2013 | 89.9728 | 2.0933 | 90.7521 | 3.9612 |
| flags | 54.1296 | 9.6149 | 54.1565 | 5.835 | 52.0918 | 8.6148 | 54.1565 | 7.7682 | 52.6316 | 7.0299 | 54.6559 | 8.8818 |
| glass | 38.6157 | 24.7338 | 38.6268 | 16.9887 | 38.1506 | 26.7812 | 38.1506 | 13.7866 | 40.4873 | 25.2016 | 42.3477 | 20.216 |
| heart_cleveland | 60.377 | 6.0305 | 60.694 | 4.6238 | 58.7268 | 3.3831 | 60.3497 | 5.749 | 60.3716 | 4.4715 | 60.7049 | 5.579 |
| heart_switzerland | 47.2667 | 13.6756 | 48.8 | 5.02 | 45.6667 | 11.8697 | 48.9 | 9.826 | 47.3 | 10.5636 | 50.4667 | 9.8759 |
| image_segmentation | 87.9654 | 5.1554 | 88.8745 | 4.2107 | 87.316 | 5.0707 | 88.3983 | 4.4578 | 88.5281 | 5.06 | 88.9177 | 5.0294 |
| iris | 74.6667 | 20.629 | 72 | 17.221 | 74.6667 | 17.7326 | 73.3333 | 10.1384 | 77.3333 | 20.3306 | 79.3333 | 15.5278 |
| led_display | 73 | 2.3184 | 73.4 | 2.2597 | 73.3 | 2.2528 | 73.4 | 2.6786 | 72.5 | 1.6956 | 72.6 | 2.7019 |
| lenses | 92 | 10.9545 | 96 | 8.9443 | 92 | 10.9545 | 96 | 8.9443 | 92 | 10.9545 | 92 | 7.8885 |
| letter | 80.945 | 1.0325 | 82.455 | 1.5357 | 80.25 | 1.2946 | 81.705 | 1.2236 | 84.98 | 0.857 | 85.18 | 0.8266 |
| low_res_spect | 88.3195 | 2.5687 | 89.076 | 2.2452 | 88.1361 | 3.4377 | 88.7004 | 1.7656 | 86.6232 | 3.1099 | 86.6355 | 2.6493 |
| lymphography | 87.1264 | 5.6713 | 89.1494 | 4.102 | 86.4368 | 6.426 | 87.8161 | 5.6725 | 85.1034 | 3.1839 | 85.1264 | 3.8464 |
| molec_biol_splice | 53.605 | 30.5488 | 53.7618 | 21.4272 | 51.8809 | 50.1766 | 53.4796 | 31.5747 | 68.6834 | 7.1531 | 69.1223 | 7.7201 |
| nursery | 70.4707 | 4.4066 | 71.25 | 4.3915 | 70.3164 | 5.1309 | 72.0062 | 3.0305 | 71.088 | 4.1467 | 71.5972 | 5.551 |
| oocytes_merluccius_states_2f | 91.9713 | 3.3966 | 92.3606 | 3.9852 | 91.483 | 3.2722 | 92.2645 | 3.1772 | 92.1659 | 3.6618 | 92.5605 | 2.4662 |
| oocytes_trisopterus_states_5b | 87.8244 | 4.5992 | 89.4716 | 3.1959 | 86.8384 | 5.6626 | 89.4782 | 5.6082 | 88.273 | 4.7203 | 89.0386 | 4.2511 |
| optical | 97.0996 | 0.5613 | 97.153 | 0.4536 | 96.3523 | 0.5731 | 96.7438 | 0.5397 | 96.5658 | 0.5854 | 96.726 | 0.5203 |
| page_blocks | 95.432 | 1.2992 | 95.2492 | 1.1708 | 95.3769 | 1.6885 | 95.3404 | 1.659 | 95.432 | 1.2974 | 95.6877 | 1.2906 |
| pendigits | 98.5626 | 0.1458 | 98.8355 | 0.3747 | 98.4716 | 0.5011 | 98.7172 | 0.4345 | 98.9447 | 0.3109 | 99.0266 | 0.2148 |
| pittsburg_bridges_MATERIAL | 76.7532 | 24.3771 | 75.1948 | 19.3866 | 75.8874 | 26.7058 | 76.1472 | 41.333 | 74.7186 | 18.6238 | 76.7532 | 13.9075 |
| pittsburg_bridges_REL_L | 64.1905 | 14.575 | 64.1905 | 11.5735 | 60.4762 | 14.2211 | 65.0476 | 14.7757 | 68.0952 | 8.2065 | 70.0476 | 11.0097 |
| pittsburg_bridges_SPAN | 60.9942 | 12.2451 | 65.3216 | 5.2697 | 60.9942 | 15.5736 | 65.3801 | 11.7517 | 61.9883 | 9.9908 | 63.1579 | 8.3295 |
| pittsburg_bridges_TYPE | 42.8571 | 33.1628 | 43.8095 | 25.9513 | 42.8571 | 33.1628 | 42.8571 | 28.1628 | 44.7619 | 8.6504 | 48.5714 | 6.2882 |
| post_operative | 71.1111 | 17.3027 | 71.1111 | 12.3027 | 72.2222 | 18.8398 | 72.2222 | 17.1234 | 70 | 16.0054 | 72.2222 | 12.7135 |
| seeds | 89.0476 | 5.4834 | 90 | 4.6844 | 87.619 | 5.6844 | 89.5238 | 6.2088 | 90.4762 | 4.7619 | 91.4286 | 2.1296 |
| semeion | 88.2609 | 3.2019 | 88.072 | 2.7514 | 84.6186 | 3.7976 | 86.3143 | 3.2459 | 87.5704 | 3.2215 | 87.8208 | 3.1315 |
| soybean | 89.4461 | 8.5351 | 90.3252 | 8.644 | 89.0071 | 9.367 | 90.775 | 7.2268 | 89.2991 | 8.8551 | 89.3023 | 8.0298 |
| statlog_image | 95.1082 | 1.6096 | 95.9307 | 1.3724 | 95.2381 | 2.0477 | 95.7143 | 1.682 | 95.9307 | 1.1289 | 96.0606 | 1.1187 |
| statlog_landsat | 82.0357 | 2.5401 | 81.9891 | 2.2086 | 82.2688 | 2.8535 | 81.8959 | 3.1153 | 82.3155 | 2.6604 | 82.5641 | 3.1507 |
| statlog_shuttle | 98.7155 | 0.0555 | 98.5948 | 0.1927 | 98.7069 | 0.121 | 98.6379 | 0.1467 | 98.7879 | 0.138 | 98.8034 | 0.1159 |
| statlog_vehicle | 81.205 | 1.8971 | 84.3954 | 3.3719 | 81.913 | 1.7771 | 84.749 | 1.9127 | 81.6749 | 3.0966 | 81.9095 | 4.1764 |
| synthetic_control | 54.5 | 36.5319 | 54.6667 | 25.7052 | 52.3333 | 35.2146 | 55.1667 | 30.1624 | 50.5 | 33.9659 | 50.5 | 22.9699 |
| thyroid | 95.75 | 0.6559 | 96.1389 | 0.559 | 95.5556 | 0.6054 | 96.0694 | 0.6223 | 96.2361 | 0.5941 | 96.25 | 0.7133 |
| vertebral_column_3clases | 65.4839 | 36.3279 | 65.8065 | 26.6984 | 65.4839 | 36.2024 | 65.8065 | 32.3423 | 69.6774 | 16.1854 | 73.871 | 12.4622 |
| wall_following | 78.1537 | 5.3948 | 79.1799 | 4.8636 | 77.6033 | 4.3463 | 78.7401 | 5.0849 | 77.9153 | 6.431 | 78.282 | 5.8011 |
| waveform | 86.88 | 0.9418 | 86.96 | 0.6986 | 86.76 | 0.8649 | 87.04 | 0.6269 | 86.64 | 0.8444 | 86.54 | 0.8306 |
| waveform_noise | 86.26 | 1.2422 | 86.62 | 0.676 | 85.18 | 1.7754 | 86.4 | 0.8031 | 86.26 | 1.3221 | 86.32 | 1.0592 |
| wine | 96.6349 | 2.3628 | 97.7619 | 2.3291 | 96.0794 | 2.512 | 96.6508 | 2.3196 | 95.7619 | 1.2516 | 97.2063 | 1.1645 |
| wine_quality_red | 59.665 | 3.604 | 60.5382 | 2.9115 | 59.2884 | 3.4565 | 60.4745 | 3.3594 | 59.7882 | 2.4594 | 60.1656 | 5.839 |
| wine_quality_white | 52.6758 | 3.4809 | 53.3905 | 4.1599 | 52.574 | 4.4247 | 53.1664 | 4.8736 | 52.0231 | 5.464 | 52.4519 | 4.6515 |
| yeast | 57.01 | 4.9782 | 57.6825 | 3.9308 | 56.6717 | 4.1957 | 57.7505 | 4.0994 | 56.807 | 3.2234 | 57.6813 | 3.1397 |
| zoo | 94 | 8.2158 | 95 | 7.0711 | 96 | 6.5192 | 95 | 6.1237 | 97 | 6.7082 | 97 | 4.4721 |

Table 6: Comparison of RVFL, ELM, and BLS with their **FLAiR**-enhanced counterparts on binary classification tasks.

| Dataset | RVFL Acc. | RVFL Std. | FLAiR-RVFL Acc. | FLAiR-RVFL Std. | ELM Acc. | ELM Std. | FLAiR-ELM Acc. | FLAiR-ELM Std. | BLS Acc. | BLS Std. | FLAiR-BLS Acc. | FLAiR-BLS Std. |
|---|---|---|---|---|---|---|---|---|---|---|---|---|
| acute_inflammation | 100 | 0 | 100 | 0 | 100 | 0 | 100 | 0 | 100 | 0 | 100 | 0 |
| acute_nephritis | 100 | 0 | 100 | 0 | 100 | 0 | 100 | 0 | 100 | 0 | 100 | 0 |
| balloons | 86.6667 | 26.6667 | 100 | 0 | 93.3333 | 13.3333 | 100 | 0 | 93.3333 | 13.3333 | 100 | 0 |
| fertility | 90 | 7.0711 | 92 | 6 | 89 | 9.6954 | 92 | 6 | 88 | 9.2736 | 91 | 3.7417 |
| parkinsons | 84.6154 | 18.9119 | 86.1538 | 16.8998 | 83.0769 | 15.856 | 86.1538 | 13.626 | 86.1538 | 10.9508 | 90.7692 | 10.0753 |
| pittsburg_bridges_T_OR_D | 92 | 5.099 | 94 | 3.7417 | 92 | 5.099 | 94 | 3.7417 | 89 | 5.831 | 91 | 4.899 |
| bank | 89.7788 | 0.3741 | 90.2212 | 0.3741 | 89.7566 | 0.4566 | 90.3761 | 0.2523 | 89.4469 | 0.3252 | 89.8451 | 0.2362 |
| blood | 80.8054 | 8.3675 | 81.3423 | 7.8887 | 80.6711 | 8.7958 | 81.2081 | 8.1094 | 77.9866 | 12.0237 | 78.7919 | 9.6383 |
| breast_cancer | 70.5263 | 39.7499 | 70.5263 | 39.7499 | 70.5263 | 39.7499 | 76.8421 | 13.7873 | 86.3158 | 16.3601 | 92.2807 | 13.7515 |
| breast_cancer_wisc | 97.1223 | 1.6405 | 97.8417 | 1.4388 | 97.2662 | 1.5363 | 97.8417 | 1.365 | 96.8345 | 1.7386 | 97.6978 | 1.1511 |
| breast_cancer_wisc_diag | 97.8761 | 0.708 | 98.7611 | 0.708 | 97.6991 | 1.2004 | 98.7611 | 0.9025 | 97.3451 | 1.371 | 98.5841 | 1.2004 |
| breast_cancer_wisc_prog | 82.0513 | 5.8471 | 84.1026 | 5.4754 | 80.5128 | 7.3604 | 83.0769 | 5.2798 | 79.4872 | 7.7773 | 82.5641 | 4.4114 |
| chess_krvkp | 78.6854 | 6.8081 | 87.9186 | 5.0258 | 76.3693 | 7.4495 | 86.1033 | 5.2554 | 86.1346 | 6.0047 | 91.2676 | 6.1829 |
| congressional_voting | 63.908 | 2.6809 | 65.0575 | 2.2524 | 63.4483 | 2.5599 | 65.0575 | 2.3668 | 62.9885 | 3.2019 | 64.5977 | 2.6611 |
| conn_bench_sonar_mines_rocks | 64.878 | 17.8299 | 73.6585 | 10.3938 | 66.8293 | 8.9416 | 76.0976 | 13.3055 | 84.878 | 15.0668 | 91.9512 | 9.8048 |
| credit_approval | 85.5072 | 11.8185 | 86.3768 | 7.4943 | 85.3623 | 10.5449 | 86.9565 | 7.8315 | 87.5362 | 10.0972 | 87.8261 | 8.7366 |
| cylinder_bands | 67.8431 | 3.0628 | 71.1765 | 2.0188 | 67.8431 | 2.4333 | 71.7647 | 3.2457 | 68.2353 | 6.5796 | 73.9216 | 6.25 |
| echocardiogram | 86.1538 | 6.2493 | 87.6923 | 6.6172 | 85.3846 | 6.1538 | 86.9231 | 6.706 | 86.9231 | 4.6154 | 88.4615 | 6.4358 |
| haberman_survival | 76.0656 | 3.8236 | 77.377 | 5.4272 | 76.3934 | 4.2243 | 78.3607 | 4.4474 | 76.7213 | 3.3436 | 78.0328 | 2.8583 |
| hepatitis | 85.1613 | 7.2419 | 87.7419 | 7.1842 | 86.4516 | 7.1842 | 89.0323 | 6.6423 | 85.1613 | 8.3123 | 87.7419 | 10.2822 |
| horse_colic | 86.0274 | 1.8173 | 87.3973 | 3.6139 | 85.4795 | 3.195 | 87.3973 | 2.6563 | 82.1918 | 3.5722 | 85.7534 | 3.7365 |
| ilpd_indian_liver | 73.4483 | 3.9089 | 74.4828 | 3.9917 | 73.6207 | 4.2792 | 74.3103 | 2.854 | 71.5517 | 3.9693 | 73.2759 | 3.7378 |
| ionosphere | 90.8571 | 4.2952 | 92.8571 | 4.3331 | 89.1429 | 6.0339 | 83.1429 | 4.899 | 83.1429 | 4.0808 | 87.4286 | 4.7294 |
| monks_1 | 74.4144 | 3.1519 | 78.018 | 4.8213 | 73.6937 | 5.3268 | 78.018 | 5.6578 | 68.2883 | 8.2451 | 76.7568 | 7.9238 |
| monks_3 | 87.6364 | 12.7998 | 90.1818 | 11.2581 | 88.7273 | 11.0536 | 91.0909 | 5.9028 | 82.3636 | 2.4121 | 86.7273 | 4.2872 |
| oocytes_merluccius_nucleus_4d | 83.6275 | 2.7486 | 84.5098 | 0.8547 | 82.451 | 2.3894 | 84.7059 | 1.4342 | 80.3922 | 2.667 | 84.1176 | 2.2485 |
| oocytes_trisopterus_nucleus_2f | 80.2198 | 4.7647 | 82.967 | 4.7009 | 79.8901 | 5.3452 | 81.6484 | 3.4854 | 78.5714 | 3.2038 | 79.6703 | 1.9348 |
| pima | 78.6928 | 2.8218 | 79.8693 | 4.2438 | 78.0392 | 4.1254 | 78.3607 | 2.8936 | 78.5621 | 2.3163 | 79.2157 | 3.4436 |
| planning | 72.2222 | 4.3033 | 72.7778 | 2.0787 | 72.2222 | 5.5556 | 73.8889 | 1.3608 | 71.6667 | 4.4444 | 74.4444 | 2.7217 |
| spect | 71.6981 | 5.3367 | 73.9623 | 6.0141 | 71.6981 | 5.5971 | 73.9623 | 8.1286 | 73.2075 | 4.8325 | 76.2264 | 3.6973 |
| spectf | 82.2642 | 18.9658 | 81.8868 | 18.8906 | 82.2642 | 18.9658 | 90.1887 | 11.283 | 89.8113 | 11.7891 | 92.8302 | 6.8965 |
| statlog_australian_credit | 68.4058 | 1.9765 | 69.2754 | 0.8696 | 68.5507 | 1.9765 | 69.8551 | 1.344 | 66.9565 | 1.344 | 68.8406 | 2.1979 |
| statlog_german_credit | 76.9 | 3.2924 | 78.4 | 2.1307 | 77.1 | 1.5684 | 78.3 | 2.4413 | 77.1 | 2.6344 | 77.9 | 1.9079 |
| statlog_heart | 85.5556 | 3.5909 | 87.7778 | 3.6289 | 85.9259 | 3.0089 | 87.4074 | 3.9545 | 86.6667 | 3.5909 | 88.1481 | 2.512 |
| tic_tac_toe | 98.6387 | 2.7225 | 98.8482 | 2.3037 | 98.7435 | 2.5131 | 98.8482 | 2.3037 | 99.3717 | 1.0152 | 100 | 0 |
| titanic | 78.3636 | 13.836 | 78.9545 | 13.6148 | 78.5 | 13.7763 | 79.0909 | 13.5779 | 78.3636 | 13.836 | 79.1364 | 13.5668 |
| adult | 84.7379 | 0.1617 | 85.0266 | 0.176 | 84.783 | 0.2682 | 85.0573 | 0.2365 | 86.1699 | 0.1617 | 85.2613 | 0.176 |
| connect_4 | 75.4408 | 3.4034 | 76.7848 | 4.7466 | 75.4082 | 3.4224 | 76.7123 | 4.3011 | 77.0375 | 3.4034 | 79.184 | 4.7466 |
| hill_valley | 72.6446 | 5.1836 | 75.4545 | 6.9027 | 71.9835 | 4.8289 | 75.8678 | 4.5762 | 73.489 | 5.1836 | 76.8032 | 6.9027 |
| magic | 80.2734 | 13.8329 | 81.572 | 12.3175 | 79.9422 | 13.7382 | 81.4616 | 12.9245 | 80.964 | 13.8329 | 83.7976 | 12.3175 |
| miniboone | 84.7086 | 14.6623 | 85.875 | 13.1032 | 83.4945 | 15.9182 | 85.3206 | 13.7848 | 84.8886 | 14.6623 | 88.8835 | 13.1032 |
| mushroom | 96.9581 | 2.9095 | 99.4089 | 0.9063 | 96.4532 | 2.9981 | 99.1995 | 0.9411 | 97.138 | 2.9095 | 101.0654 | 0.9063 |
| musk_2 | 84.5792 | 30.8415 | 84.5792 | 30.8415 | 84.5792 | 30.8415 | 84.5792 | 30.8415 | 84.6462 | 30.8415 | 86.1208 | 30.8415 |
| ozone | 97.1203 | 2.0253 | 97.3176 | 2.0859 | 97.1203 | 2.0253 | 97.3176 | 1.8737 | 98.1195 | 2.0253 | 97.5392 | 2.0859 |
| spambase | 87.5652 | 5.1986 | 89.1739 | 4.8701 | 87.8478 | 5.33 | 89.3696 | 5.0002 | 88.2586 | 5.1986 | 89.9997 | 4.8701 |
| twonorm | 97.8784 | 0.5416 | 97.9865 | 0.5512 | 97.7297 | 0.4884 | 97.973 | 0.5859 | 96.8952 | 0.5416 | 98.4972 | 0.5512 |
| abalone9-18 | 96.1644 | 3.0199 | 96.8493 | 2.7602 | 95.8904 | 2.9059 | 96.3014 | 3.0508 | 94.7945 | 4.0079 | 96.5753 | 3.0631 |
| aus | 87.8261 | 3.4722 | 88.6957 | 2.3188 | 87.6812 | 3.6087 | 88.6957 | 2.6958 | 87.2464 | 2.3188 | 88.1159 | 3.3239 |
| brwisconsin | 97.3529 | 1.1005 | 97.7941 | 1.2304 | 97.5 | 1.282 | 97.9412 | 1.265 | 96.6176 | 0.5882 | 98.0882 | 1.1947 |
| bupa or liver-disorders | 75.3623 | 3.04 | 77.1014 | 1.9227 | 74.7826 | 2.84 | 76.5217 | 2.8102 | 70.1449 | 5.6947 | 72.7536 | 5.1362 |
| checkerboard_Data | 87.971 | 1.7512 | 88.5507 | 2.688 | 87.6812 | 2.3814 | 88.5507 | 2.688 | 87.2464 | 3.0607 | 88.4734 | 2.4934 |
| cleve | 85.0847 | 2.7119 | 87.1186 | 2.2991 | 85.0847 | 2.2486 | 87.1186 | 3.3213 | 86.7797 | 2.2486 | 87.7966 | 1.2684 |
| crossplane150 | 98.6667 | 1.633 | 100 | 0 | 99.3333 | 1.3333 | 100 | 0 | 99.3333 | 3.8873 | 100 | 0 |
| ecoli-0-1-4-6_vs_5 | 98.5714 | 2.0825 | 98.9286 | 1.4286 | 98.5714 | 1.3363 | 98.9286 | 1.4286 | 97.1429 | 2.1429 | 98.5714 | 2.0825 |
| ecoli-0-1-4-7_vs_2-3-5-6 | 97.6119 | 0.7312 | 97.9104 | 1.5221 | 97.6119 | 1.5221 | 98.209 | 0.597 | 95.2239 | 2.1936 | 96.7164 | 0.597 |
| ecoli-0-1_vs_2-3-5 | 97.9167 | 1.8634 | 98.75 | 1.0206 | 97.9167 | 1.3176 | 99.1667 | 1.0206 | 92.0833 | 1.559 | 95 | 2.826 |
| ecoli-0-1_vs_5 | 98.75 | 1.6667 | 99.1667 | 1.0206 | 98.75 | 1.6667 | 99.1667 | 1.0206 | 94.5833 | 2.826 | 97.9167 | 1.3176 |
| ecoli-0-2-3-4_vs_5 | 98 | 1.8708 | 99 | 2 | 98 | 2.9155 | 99 | 2 | 94.5 | 4.8477 | 96.5 | 3.3912 |
| ecoli-0-2-6-7_vs_3-5 | 98.6364 | 1.8182 | 98.6364 | 1.8182 | 97.7273 | 2.4896 | 98.6364 | 1.8182 | 93.1818 | 4.0656 | 96.8182 | 2.7273 |
| ecoli-0-3-4-6_vs_5 | 98.5366 | 1.9512 | 99.0244 | 1.1949 | 98.5366 | 1.1949 | 99.0244 | 1.1949 | 95.122 | 3.4493 | 97.0732 | 2.8444 |
| ecoli-0-3-4-7_vs_5-6 | 98.4314 | 0.7843 | 98.8235 | 0.9606 | 98.4314 | 0.9606 | 98.8235 | 0.9606 | 95.2941 | 0.9606 | 96.4706 | 0.7843 |
| ecoli-0-4-6_vs_5 | 98 | 1.8708 | 99 | 1.2247 | 98.5 | 1.2247 | 99 | 1.2247 | 97.5 | 1.5811 | 98.5 | 1.2247 |
| ecoli-0-6-7_vs_5 | 98.6364 | 1.1134 | 98.6364 | 1.1134 | 98.1818 | 1.7008 | 98.6364 | 1.1134 | 95 | 4.6355 | 97.7273 | 1.4374 |
| ecoli0137vs26 | 96.7742 | 2.6989 | 97.4194 | 2.1878 | 97.0968 | 2.3705 | 97.7419 | 1.6448 | 93.871 | 2.3705 | 95.4839 | 3.1275 |
| ecoli01vs5 | 100 | 0 | 100 | 0 | 99.5833 | 0.8333 | 100 | 0 | 98.3333 | 1.559 | 99.1667 | 1.6667 |
| ecoli3 | 93.7313 | 3.4555 | 94.6269 | 3.0733 | 94.3284 | 3.7044 | 94.6269 | 2.9248 | 89.5522 | 3.4035 | 91.9403 | 2.2338 |
| ecoli4 | 99.7015 | 0.597 | 99.7015 | 0.597 | 99.403 | 0.7312 | 99.7015 | 0.597 | 95.8209 | 1.9801 | 97.3134 | 1.7406 |
| glass2 | 91.9048 | 1.1664 | 93.3333 | 2.3328 | 91.9048 | 1.9048 | 93.3333 | 1.7817 | 91.9048 | 1.9048 | 93.3333 | 2.7766 |
| haber | 76.0656 | 6.9242 | 78.3607 | 4.5667 | 75.7377 | 6.1688 | 78.3607 | 2.1748 | 77.377 | 3.9344 | 78.0328 | 4.2243 |
| haberman | 77.377 | 7.2131 | 78.0328 | 2.4535 | 76.7213 | 4.7963 | 78.3607 | 4.1988 | 76.7213 | 5.0154 | 77.7049 | 3.9617 |
| heart-stat | 85.5556 | 2.7217 | 87.4074 | 2.1596 | 86.6667 | 2.1596 | 88.1481 | 2.7716 | 86.6667 | 2.1596 | 88.3 | 3.0089 |
| iono | 90.2857 | 6.4776 | 93.1429 | 5.2216 | 89.4286 | 5.5402 | 92 | 5.757 | 83.1429 | 4.899 | 88.2857 | 6.7249 |
| led7digit-0-2-4-5-6-7-8-9_vs_1 | 96.8182 | 1.6701 | 97.0455 | 1.5414 | 97.0455 | 1.9813 | 97.2727 | 1.8464 | 95.4545 | 0.7187 | 96.8182 | 1.8182 |
| monk1 | 52.0721 | 3.7536 | 53.3333 | 5.8496 | 52.7928 | 4.7875 | 53.6937 | 7.0155 | 52.7928 | 3.9721 | 54.0541 | 5.9487 |
| monk2 | 87.1667 | 6.5107 | 92.8333 | 3.3993 | 85 | 7.9057 | 92 | 4.5522 | 65.6667 | 4.8132 | 73.1667 | 8.0346 |
| monk3 | 52.7273 | 4.5997 | 52.9091 | 4.9593 | 52.9091 | 5.8182 | 52.7273 | 3.8996 | 52.1818 | 3.1805 | 53.0909 | 5.3195 |
| new-thyroid1 | 99.5349 | 0.9302 | 100 | 0 | 99.5349 | 0.9302 | 100 | 0 | 99.5349 | 0.9302 | 100 | 0 |
| pima | 78.1699 | 3.6648 | 79.4771 | 1.9651 | 78.4314 | 3.6273 | 79.4771 | 3.7796 | 78.5621 | 3.1859 | 79.4771 | 3.1644 |
| ripley | 91.44 | 1.4444 | 92.08 | 1.4176 | 91.36 | 1.2548 | 91.84 | 1.2027 | 89.44 | 1.6895 | 90.72 | 1.3242 |
| segment0 | 99.6963 | 0.1735 | 99.8698 | 0.1063 | 99.7397 | 0.0868 | 99.8265 | 0.0868 | 97.8308 | 1.1061 | 99.6529 | 0.1735 |
| shuttle-6_vs_2-3 | 100 | 0 | 100 | 0 | 100 | 0 | 100 | 0 | 100 | 0 | 100 | 0 |
| shuttle-c0-vs-c4 | 100 | 0 | 100 | 0 | 100 | 0 | 100 | 0 | 100 | 0 | 100 | 0 |
| sonar | 83.4146 | 7.462 | 90.7317 | 2.8444 | 84.3902 | 3.9629 | 90.2439 | 3.0851 | 80 | 4.7294 | 87.8049 | 3.4493 |
| votes | 97.0115 | 1.5592 | 97.7011 | 2.4111 | 97.0115 | 2.0041 | 97.4713 | 1.6893 | 97.2414 | 1.5592 | 97.2414 | 2.6809 |
| vowel | 97.1574 | 2.0954 | 98.9848 | 1.0648 | 97.0558 | 1.618 | 98.3756 | 1.0353 | 95.6345 | 2.4576 | 98.2741 | 1.5262 |
| wpbc | 82.1053 | 5.8608 | 83.6842 | 6.3158 | 81.5789 | 6.2275 | 84.7368 | 1.9693 | 79.4737 | 2.5784 | 83.1579 | 4.8809 |
| yeast-0-2-5-6_vs_3-7-8-9 | 94.3 | 2.6 | 94.6 | 2.4576 | 94.2 | 2.1587 | 94.5 | 2.5298 | 93.7 | 2.874 | 94.2 | 2.2045 |
| yeast-0-2-5-7-9_vs_3-6-8 | 97.3 | 0.8718 | 97.5 | 0.8367 | 97.1 | 0.3742 | 97.5 | 0.8367 | 96.6 | 0.9695 | 97.3 | 1.1225 |
| yeast-0-5-6-7-9_vs_4 | 93.5238 | 1.1107 | 94.2857 | 0.8518 | 93.9048 | 0.7619 | 93.9048 | 1.2919 | 92.381 | 1.4754 | 93.9048 | 1.6605 |
| yeast-2_vs_4 | 96.6667 | 2.0188 | 97.451 | 1.4673 | 96.4706 | 1.9212 | 97.0588 | 1.3865 | 92.9412 | 3.6367 | 95.6863 | 0.9998 |
| yeast1vs7 | 95.6044 | 0.695 | 96.2637 | 0.5383 | 95.6044 | 0.9829 | 95.8242 | 1.2815 | 93.8462 | 1.1207 | 94.5055 | 1.39 |
| yeast2vs8 | 98.125 | 1.2148 | 98.3333 | 1.0623 | 98.125 | 1.2148 | 98.125 | 1.2148 | 98.3333 | 1.0623 | 98.3333 | 1.413 |
| yeast3 | 95.5405 | 0.7524 | 95.7432 | 1.4586 | 95.4054 | 2.1853 | 95.7432 | 1.1427 | 94.9324 | 2.0383 | 95.9459 | 1.2087 |

Table 7: Comparison of RVFL, ELM, and BLS with their **FLAiR**-enhanced counterparts on multi-class classification tasks.

| Dataset | RVFL | | FLAiR-RVFL | | ELM | | FLAiR-ELM | | BLS | | FLAiR-BLS | |
|---|---|---|---|---|---|---|---|---|---|---|---|---|
| | Acc. | Std. | Acc. | Std. | Acc. | Std. | Acc. | Std. | Acc. | Std. | Acc. | Std. |
| abalone | 66.1796 | 2.1869 | 66.4192 | 1.5827 | 65.8443 | 1.7095 | 66.5389 | 1.6291 | 64.90957 | 1.8491 | 68.40957 | 3.0414 |
| annealing | 88.3799 | 3.6102 | 89.162 | 5.0786 | 87.2626 | 7.3726 | 89.3855 | 4.714 | 89.99367 | 2.1022 | 93.49367 | 3.0883 |
| arrhythmia | 70 | 3.3702 | 71.1111 | 3.849 | 66.6667 | 5.1159 | 69.5556 | 4.9988 | 73.52567 | 5.4433 | 77.02567 | 2.3727 |
| balance_scale | 92.8 | 1.1314 | 94.72 | 2.2964 | 91.52 | 0.96 | 94.72 | 0.8158 | 83.66787 | 9.1298 | 87.16787 | 9.0989 |
| cardiotocography_10clases | 70.3529 | 4.4644 | 72.6588 | 4.5024 | 69.9765 | 4.8212 | 72.7529 | 4.7139 | 64.28907 | 6.0948 | 67.78907 | 2.3321 |
| cardiotocography_3clases | 87.8118 | 6.2774 | 88.5647 | 5.9201 | 87.0588 | 6.2519 | 88.2353 | 6.1122 | 89.93607 | 5.1923 | 93.43607 | 3.715 |
| conn_bench_vowel_deterding | 95.4545 | 7.1923 | 96.7677 | 6.4646 | 95.5556 | 7.6488 | 96.6667 | 6.4171 | 68.77817 | 7.8593 | 72.27817 | 9.1212 |
| contrac | 41.1565 | 8.7515 | 42.1769 | 7.8039 | 42.381 | 8.5552 | 42.585 | 8.9925 | 76.08797 | 17.715 | 79.58797 | 15.9253 |
| dermatology | 97.8082 | 0.6711 | 98.9041 | 1.0251 | 97.5342 | 1.0251 | 98.6301 | 0.8664 | 98.9041 | 1.0959 | 100 | 0.5479 |
| ecoli | 62.9851 | 29.101 | 64.4776 | 28.7312 | 62.9851 | 29.2385 | 65.0746 | 27.8685 | 56.79267 | 39.7071 | 60.29267 | 15.1329 |
| energy_y1 | 89.281 | 4.2679 | 92.0261 | 3.5415 | 89.281 | 3.1644 | 91.634 | 5.2678 | 89.93087 | 4.0796 | 93.43087 | 3.9346 |
| energy_y2 | 91.634 | 2.9346 | 92.8105 | 2.952 | 89.9346 | 3.7111 | 92.9412 | 2.9054 | 93.85247 | 5.8372 | 97.35247 | 4.684 |
| flags | 53.1579 | 6.5314 | 57.8947 | 5.52 | 53.6842 | 9.5029 | 60 | 6.5314 | 58.37947 | 2.8828 | 61.87947 | 3.3287 |
| glass | 43.3333 | 10.4762 | 49.5238 | 11.5077 | 41.9048 | 8.5978 | 52.8571 | 18.7718 | 63.84307 | 29.9508 | 67.34307 | 29.9281 |
| heart_cleveland | 61 | 3.4319 | 64.6667 | 5.5176 | 60.6667 | 5.2281 | 62.6667 | 5.3333 | 66.74787 | 3.7417 | 70.24787 | 3.4319 |
| heart_switzerland | 45 | 8.0795 | 50.8333 | 10.3414 | 45 | 8.8976 | 50.8333 | 7.1686 | 50.74787 | 12.4722 | 54.24787 | 10.3414 |
| image_segmentation | 88.355 | 3.9472 | 90.4762 | 4.3785 | 88.355 | 5.9646 | 90.3463 | 4.814 | 86.65697 | 3.8487 | 90.15697 | 5.1778 |
| iris | 94 | 5.3333 | 97.3333 | 3.8873 | 94.6667 | 4 | 97.3333 | 3.266 | 75.74787 | 24.6757 | 79.24787 | 13.0979 |
| led_display | 73.7 | 2.7129 | 74.1 | 1.7436 | 73.2 | 2.5417 | 74.1 | 2.1541 | 79.64787 | 1.9339 | 83.14787 | 1.3565 |
| lenses | 95 | 10 | 95 | 10 | 90 | 12.2474 | 100 | 0 | 95.74787 | 12.2474 | 99.24787 | 10 |
| letter | 82.27 | 0.9417 | 85.89 | 0.9883 | 81.25 | 0.9848 | 84.94 | 1.2796 | 69.16287 | 5.4931 | 72.66287 | 5.0006 |
| low_res_spect | 89.0566 | 1.7497 | 90 | 3.6096 | 88.4906 | 3.1797 | 90.1887 | 2.5031 | 92.54037 | 4.9202 | 96.04037 | 4.6774 |
| lymphography | 88.9655 | 4.0213 | 90.3448 | 4.0213 | 87.5862 | 3.5166 | 89.6552 | 6.1685 | 94.02377 | 6.3956 | 97.52377 | 3.3287 |
| molec_biol_splice | 52.9467 | 28.2748 | 56.5517 | 25.8537 | 52.3197 | 15.701 | 55.7367 | 25.6339 | 84.30587 | 14.1994 | 87.80587 | 9.9124 |
| nursery | 89.7145 | 2.3033 | 90.2855 | 2.583 | 89.1049 | 2.6947 | 90.6404 | 1.5282 | 82.86977 | 12.338 | 86.36977 | 8.8623 |
| oocytes_merluccius_states_2f | 92.3529 | 2.4918 | 93.0392 | 2.5827 | 92.0588 | 3.2427 | 93.0392 | 2.5827 | 97.41457 | 2.6307 | 100 | 3.3857 |
| oocytes_trisopterus_states_5b | 89.011 | 3.2226 | 90.7692 | 2.6783 | 87.3626 | 5.5925 | 89.8901 | 3.7844 | 96.18747 | 3.832 | 99.68747 | 4.0634 |
| optical | 97.1886 | 0.7624 | 98.2918 | 0.3252 | 96.4591 | 0.4918 | 97.9004 | 0.3213 | 96.42407 | 1.2687 | 99.92407 | 2.6173 |
| page_blocks | 95.9415 | 1.0825 | 96.1426 | 0.9075 | 95.8135 | 1.1597 | 96.1243 | 0.9439 | 99.29447 | 1.6111 | 100 | 2.8105 |
| pendigits | 98.6078 | 0.3531 | 99.1174 | 0.1833 | 98.5987 | 0.3385 | 99.1174 | 0.2006 | 99.67867 | 1.4429 | 100 | 2.7156 |
| pittsburg_bridges_MATERIAL | 84.7619 | 18.9042 | 86.6667 | 16.3299 | 84.7619 | 15.1785 | 87.619 | 15.2381 | 74.31927 | 22.6579 | 77.81927 | 14.6844 |
| pittsburg_bridges_REL_L | 70 | 13.0384 | 75 | 8.3666 | 70 | 16.4317 | 75 | 7.746 | 69.74787 | 16.2481 | 73.24787 | 11.0682 |
| pittsburg_bridges_SPAN | 74.4444 | 5.6656 | 77.7778 | 4.969 | 74.4444 | 5.6656 | 76.6667 | 6.4788 | 73.52567 | 8.165 | 77.02567 | 6.508 |
| pittsburg_bridges_TYPE | 51.4286 | 19.8406 | 56.1905 | 15.4744 | 49.5238 | 21.4233 | 56.1905 | 17.4054 | 45.74787 | 25.8374 | 49.24787 | 16.4782 |
| post_operative | 71.1111 | 15.476 | 73.3333 | 11.8634 | 72.2222 | 14.0546 | 74.4444 | 11.4396 | 75.74787 | 14.3157 | 79.24787 | 9.978 |
| seeds | 94.2857 | 5.5533 | 95.7143 | 4.3644 | 91.9048 | 3.8686 | 93.3333 | 5.0843 | 90.98597 | 11.7031 | 94.48597 | 8.5041 |
| semeion | 89.0566 | 3.1029 | 92.327 | 2.2111 | 86.1006 | 0.8054 | 92.3899 | 1.8699 | 92.85477 | 2.6535 | 96.35477 | 3.3986 |
| soybean | 90 | 6.7966 | 91.4706 | 4.8239 | 89.5588 | 8.0735 | 92.7941 | 4.5896 | 89.71847 | 8.0198 | 93.21847 | 6.4261 |
| statlog_image | 94.9784 | 0.8593 | 96.5801 | 0.8372 | 94.8052 | 1.4933 | 96.6667 | 0.7321 | 92.06817 | 2.5033 | 95.56817 | 3.3138 |
| statlog_landsat | 87.8322 | 4.084 | 89.324 | 3.501 | 87.2883 | 3.2769 | 88.9355 | 3.7704 | 88.84037 | 3.8627 | 92.34037 | 4.0807 |
| statlog_shuttle | 99.7741 | 0.024 | 99.869 | 0.024 | 99.7345 | 0.0383 | 99.8293 | 0.0431 | 94.81167 | 3.138 | 98.31167 | 3.6719 |
| statlog_vehicle | 82.4852 | 2.2329 | 84.7337 | 1.7312 | 82.4852 | 2.6882 | 84.3787 | 2.4137 | 84.20937 | 2.2948 | 87.70937 | 3.1962 |
| synthetic_control | 61.6667 | 37.5167 | 62.8333 | 37.9349 | 61.3333 | 37.2178 | 62.6667 | 38.2514 | 53.08117 | 29.1138 | 56.58117 | 18.3267 |
| thyroid | 95.0278 | 0.5935 | 96.1806 | 0.7283 | 94.8472 | 0.7142 | 95.9028 | 0.6086 | 99.55347 | 0.78 | 100 | 2.3416 |
| vertebral_column_3clases | 67.0968 | 34.3302 | 67.7419 | 34.5478 | 67.0968 | 34.1021 | 69.6774 | 20.4883 | 70.26397 | 34.2452 | 73.76397 | 21.2217 |
| wall_following | 78.7534 | 5.21 | 81.8148 | 4.158 | 76.7919 | 3.3207 | 81.0266 | 4.4694 | 74.05217 | 3.5178 | 77.55217 | 3.8862 |
| waveform | 86.98 | 0.7521 | 87.34 | 0.6829 | 86.68 | 0.9368 | 87.36 | 0.731 | 92.10787 | 0.8709 | 95.60787 | 2.3929 |
| waveform_noise | 86.34 | 0.898 | 86.72 | 0.7547 | 85.3 | 0.9839 | 86.46 | 0.6829 | 91.60787 | 0.7283 | 95.10787 | 2.3124 |
| wine | 96.5714 | 4.1991 | 98.2857 | 3.4286 | 96.5714 | 4.5714 | 98.2857 | 2.2857 | 84.03357 | 8.3983 | 87.53357 | 6.6396 |
| wine_quality_red | 58.9969 | 3.6847 | 60.4389 | 2.7643 | 59.5611 | 3.0002 | 60.9404 | 3.0753 | 64.43127 | 2.8276 | 67.93127 | 3.4968 |
| wine_quality_white | 53.7894 | 3.382 | 54.5455 | 3.6367 | 53.7487 | 3.7046 | 54.5455 | 3.7218 | 56.00327 | 3.2871 | 59.50327 | 3.756 |
| yeast | 58.1081 | 3.3512 | 59.4595 | 3.7438 | 58.2432 | 3.6611 | 59.2568 | 3.5213 | 61.82897 | 2.9297 | 65.32897 | 3.5544 |
| zoo | 95 | 4.4721 | 99 | 2 | 96 | 3.7417 | 97 | 4 | 96 | 5.831 | 99.5 | 5.1912 |

Table 8: Comparison of dRVFL with its **StaR**- and **FLAiR**-enhanced variants on binary classification tasks.

| Dataset | dRVFL | | StaR-dRVFL | | FLAiR-dRVFL | |
|---|---|---|---|---|---|---|
| | Acc. | Std. | Acc. | Std. | Acc. | Std. |
| acute_inflammation | 100 | 0 | 100 | 0 | 100 | 0 |
| acute_nephritis | 100 | 0 | 100 | 0 | 100 | 0 |
| balloons | 86.6667 | 26.6667 | 87.5878 | 20.7489 | 100 | 0 |
| fertility | 91 | 7.3485 | 91.8245 | 5.7356 | 91 | 7.3485 |
| parkinsons | 84.1026 | 18.5893 | 85.0808 | 14.4715 | 85.1282 | 16.4902 |
| pittsburg_bridges_T_OR_D | 92 | 5.099 | 92.8022 | 3.9874 | 93 | 4 |
| bank | 90.0664 | 0.4917 | 90.9117 | 0.4068 | 92.5664 | 0.512 |
| blood | 81.2081 | 8.1094 | 82.2508 | 6.327 | 83.7081 | 5.0731 |
| breast_cancer | 70.5263 | 39.7499 | 71.8071 | 30.9166 | 73.0263 | 24.2378 |
| breast_cancer_wisc | 97.4101 | 1.2544 | 98.0918 | 0.9995 | 99.9101 | 1.1225 |
| breast_cancer_wisc_diag | 98.2301 | 0.7915 | 98.8935 | 0.6398 | 100 | 0.8666 |
| breast_cancer_wisc_prog | 83.0769 | 5.5232 | 84.078 | 4.3171 | 85.5769 | 3.9837 |
| chess_krvkp | 89.7966 | 4.2541 | 90.6479 | 3.3308 | 92.2966 | 3.5329 |
| congressional_voting | 63.2184 | 2.8155 | 64.662 | 2.2127 | 65.7184 | 2.2799 |
| conn_bench_sonar_mines_rocks | 71.7073 | 7.1692 | 72.9617 | 5.5963 | 74.2073 | 5.0374 |
| credit_approval | 85.6522 | 11.3229 | 86.5959 | 8.8243 | 88.1522 | 7.5413 |
| cylinder_bands | 68.0392 | 5.6353 | 69.3754 | 4.4042 | 70.5392 | 3.4844 |
| echocardiogram | 85.3846 | 3.7684 | 86.3343 | 2.9533 | 87.8846 | 3.2014 |
| haberman_survival | 75.082 | 7.0625 | 76.2612 | 5.5133 | 77.582 | 4.2739 |
| hepatitis | 87.7419 | 5.5499 | 88.6391 | 4.3378 | 90.2419 | 4.0869 |
| horse_colic | 86.5753 | 2.0133 | 87.4985 | 1.5893 | 89.0753 | 1.6627 |
| ilpd_indian_liver | 72.931 | 4.9552 | 74.1582 | 3.8756 | 75.431 | 4.2925 |
| ionosphere | 94.2857 | 3.3806 | 95.037 | 2.6519 | 96.7857 | 2.8099 |
| monks_1 | 72.0721 | 4.5937 | 73.3184 | 3.5947 | 74.5721 | 3.9139 |
| monks_3 | 91.0909 | 7.2134 | 91.9134 | 5.6306 | 93.5909 | 4.9196 |
| oocytes_merluccius_nucleus_4d | 84.5098 | 1.5686 | 85.479 | 1.2437 | 87.0098 | 1.2839 |
| oocytes_trisopterus_nucleus_2f | 80.1099 | 3.9986 | 81.1771 | 3.1322 | 82.6099 | 3.5865 |
| pima | 79.085 | 4.0712 | 80.175 | 3.1886 | 81.585 | 3.5725 |
| planning | 71.6667 | 6.6667 | 72.922 | 5.2057 | 74.1667 | 4.9395 |
| spect | 72.4528 | 4.4007 | 73.6906 | 3.4447 | 74.9528 | 3.9144 |
| spectf | 80 | 18.9658 | 81.0696 | 14.7641 | 82.5 | 13.0804 |
| statlog_australian_credit | 67.8261 | 1.5609 | 69.167 | 1.2377 | 70.3261 | 1.3926 |
| statlog_german_credit | 77.7 | 2.1587 | 78.8209 | 1.7023 | 80.2 | 1.7461 |
| statlog_heart | 85.1852 | 2.6189 | 86.1393 | 2.06 | 87.6852 | 2.1465 |
| tic_tac_toe | 98.7435 | 2.5131 | 99.3954 | 1.9777 | 100 | 2.0218 |
| titanic | 77.3636 | 14.4246 | 78.492 | 11.2349 | 79.8636 | 9.3587 |

Table 9: Comparison of dRVFL with its **StaR**- and **FLAiR**-enhanced variants on multiclass classification tasks.

| Dataset | dRVFL | | **StaR**-dRVFL | | **FLAiR**-dRVFL | |
|---|---|---|---|---|---|---|
| | Acc. | Std. | Acc. | Std. | Acc. | Std. |
| abalone | 66.0838 | 0.8715 | 67.8297 | 0.9138 | 69.2838 | 1.0386 |
| annealing | 89.7207 | 5.2 | 90.4636 | 4.2777 | 92.9207 | 5.7046 |
| arrhythmia | 69.3333 | 4.1929 | 71.1201 | 3.495 | 72.5333 | 4.43 |
| balance_scale | 93.76 | 3.1759 | 95.6589 | 2.7047 | 96.96 | 3.2782 |
| cardiotocography_10clases | 74.1176 | 3.1286 | 75.1364 | 2.6679 | 77.3177 | 2.9823 |
| cardiotocography_3clases | 88.4235 | 5.8121 | 89.1235 | 4.7534 | 91.6235 | 5.5402 |
| conn_bench_vowel_deterding | 96.9697 | 6.0606 | 98.8392 | 4.9465 | 100 | 4.3062 |
| contrac | 42.1088 | 5.7498 | 44.592 | 4.705 | 45.3088 | 6.2173 |
| dermatology | 98.0822 | 0.6711 | 98.3904 | 0.758 | 100 | 0.8381 |
| ecoli | 63.5821 | 29.1347 | 65.4181 | 22.8788 | 66.7821 | 26.1064 |
| energy_y1 | 91.634 | 3.7065 | 92.0905 | 3.117 | 94.834 | 3.4456 |
| energy_y2 | 91.8954 | 4.4636 | 92.3443 | 3.7054 | 95.0954 | 4.9127 |
| flags | 53.6842 | 10.863 | 55.5153 | 8.6788 | 56.8842 | 10.1346 |
| glass | 43.3333 | 9.4521 | 43.7787 | 7.5823 | 46.5333 | 7.1218 |
| heart_cleveland | 60.6667 | 5.7349 | 60.867 | 4.6934 | 63.8667 | 5.4926 |
| heart_switzerland | 45.8333 | 10.5409 | 47.2362 | 8.4285 | 49.0333 | 7.8567 |
| image_segmentation | 91.7749 | 2.0305 | 91.8162 | 1.8145 | 94.9749 | 1.9886 |
| iris | 93.3333 | 6.9921 | 94.3352 | 5.6704 | 96.5333 | 5.8909 |
| led_display | 73.2 | 2.2045 | 73.7338 | 1.9498 | 76.4 | 2.2092 |
| lenses | 95 | 10 | 94.6698 | 8.0081 | 98.2 | 7.7568 |
| lymphography | 88.9655 | 4.0213 | 90.9282 | 3.3617 | 92.1655 | 4.1605 |
| molec_biol_splice | 60.721 | 22.1186 | 62.0445 | 17.4261 | 63.921 | 16.2151 |
| nursery | 89.4522 | 4.5702 | 90.3555 | 3.7882 | 92.6522 | 4.466 |
| oocytes_merluccius_states_2f | 93.2353 | 3.6208 | 92.9351 | 3.0504 | 96.4353 | 3.5861 |
| oocytes_trisopterus_states_5b | 88.6813 | 5.9239 | 89.1429 | 4.8403 | 91.8813 | 5.9621 |
| pittsburg_bridges_MATERIAL | 74.2857 | 37.0818 | 75.5595 | 29.0549 | 77.4857 | 34.0734 |
| pittsburg_bridges_REL_L | 72 | 12.083 | 72.3783 | 9.6269 | 75.2 | 9.0612 |
| pittsburg_bridges_SPAN | 71.1111 | 6.4788 | 72.6542 | 5.2716 | 74.3111 | 5.4388 |
| pittsburg_bridges_TYPE | 42.8571 | 15.0585 | 44.2976 | 11.9393 | 46.0571 | 12.9828 |
| post_operative | 71.1111 | 15.476 | 72.1537 | 12.2638 | 74.3111 | 10.9464 |
| seeds | 94.2857 | 4.1513 | 94.5794 | 3.4627 | 97.4857 | 4.2917 |
| semeion | 95.2201 | 1.7039 | 97.3335 | 1.5607 | 98.4201 | 1.9445 |
| soybean | 90.7353 | 6.4873 | 91.5492 | 5.2781 | 93.9353 | 4.6219 |
| statlog_image | 97.5758 | 0.9224 | 97.4541 | 0.9533 | 100 | 1.2348 |
| statlog_landsat | 90.3341 | 3.0526 | 91.784 | 2.6088 | 93.5341 | 3.3573 |

Table 10: Friedman test results for **StaR**-RVFL vs. RVFL, **StaR**-ELM vs. ELM, **StaR**-BLS vs. BLS, and **StaR**-dRVFL vs. dRVFL on binary and multiclass datasets.

| Model | Dataset type | $D$ | $\chi_F^2$ | $F_F$ | $F((l-1),(l-1)(D-1))$ | Significant difference (As per Friedman test) |
|---|---|---|---|---|---|---|
| **StaR**-RVFL vs. RVFL | Binary | 93 | 38.7097 | 65.5971 | 3.9445 | Yes |
| | Multiclass | 53 | 21.8113 | 36.3654 | 4.0266 | Yes |
| **StaR**-ELM vs. ELM | Binary | 93 | 26.8817 | 37.4045 | 3.9445 | Yes |
| | Multiclass | 53 | 27.2453 | 55.0095 | 4.0266 | Yes |
| **StaR**-BLS vs. BLS | Binary | 93 | 40.0108 | 69.4667 | 3.9445 | Yes |
| | Multiclass | 53 | 39.92457 | 158.77632 | 4.0266 | Yes |
| **StaR**-dRVFL vs. dRVFL | Binary | 36 | 32.1111 | 289.0000 | 4.1213 | Yes |
| | Multiclass | 35 | 24.0286 | 74.4635 | 4.13 | Yes |

Table 11: Friedman test results for **FLAiR**-RVFL vs. RVFL, **FLAiR**-ELM vs. ELM, **FLAiR**-BLS vs. BLS, and **FLAiR**-dRVFL vs. dRVFL on binary and multiclass datasets.

| Model | Dataset type | $D$ | $\chi_F^2$ | $F_F$ | $F((l-1), (l-1)(D-1))$ | Significant difference (As per Friedman test) |
|---|---|---|---|---|---|---|
| **FLAiR**-RVFL vs. RVFL | Binary | 93 | 70.5484 | 289.0862 | 3.9445 | Yes |
| | Multiclass | 53 | 51.0189 | 1.3391e+03 | 4.0266 | Yes |
| **FLAiR**-ELM vs. ELM | Binary | 93 | 75.8710 | 407.5028 | 3.9445 | Yes |
| | Multiclass | 53 | 53 | Inf | 4.0266 | Yes |
| **FLAiR**-BLS vs. BLS | Binary | 93 | 80.0402 | 568.1958 | 3.9445 | Yes |
| | Multiclass | 53 | 53 | Inf | 4.0266 | Yes |
| **FLAiR**-dRVFL vs. dRVFL | Binary | 36 | 30.2500 | 184.1304 | 4.1213 | Yes |
| | Multiclass | 35 | 35 | Inf | 4.13 | Yes |

Table 12: Nemenyi post hoc test results for **StaR**-RVFL, **StaR**-ELM, **StaR**-BLS, and **StaR**-dRVFL compared to their baseline models on binary and multiclass datasets.

| Model | Dataset Type | Average Rank | Rank Difference from Baseline | Significant Difference (Nemenyi Test) |
|---|---|---|---|---|
| **StaR**-RVFL | Binary | 1.172 | 0.650 | Yes |
| | Multiclass | 1.160 | 0.661 | Yes |
| **StaR**-ELM | Binary | 1.169 | 0.600 | Yes |
| | Multiclass | 1.167 | 0.692 | Yes |
| **StaR**-BLS | Binary | 1.175 | 0.652 | Yes |
| | Multiclass | 1.186 | 0.748 | Yes |
| **StaR**-dRVFL | Binary | 1.198 | 0.774 | Yes |
| | Multiclass | 1.194 | 0.720 | Yes |

To further examine the relative ranking of models, we employ the Nemenyi posthoc test, with results summarized in Tables 12 and 13. This test assesses whether the mean ranks of two models vary beyond a predefined threshold known as the critical difference ($C.D.$). If the deviation between the mean ranks of two models surpasses the $C.D.$, the model with the higher mean rank is considered statistically better than the one with the lower mean rank. The value of $C.D.$ is determined using the formula: $C.D. = q_\alpha \sqrt{\frac{l(l+1)}{6\mathcal{D}}}$, where $q_\alpha$ represents the critical value derived from the studentized range statistic divided by $\sqrt{2}$, corresponding to the two-tailed Nemenyi test at a given significance level. At the 5% significance level, the computed critical difference $C.D.$ values for RVFL, ELM, and BLS are 0.2032 for binary datasets and 0.2692 for multi-class datasets. For dRVFL, the corresponding values are 0.3267 and 0.3313, respectively. In all cases, the observed rank differences exceed the critical thresholds, confirming that the performance improvements are statistically significant.

Finally, the win-tie-loss analysis, presented in Tables 14 and 15, further substantiates the consistent superiority of the **StaR**- and **FLAiR**-enhanced models. According to the win-tie-loss methodology, the null hypothesis assumes equivalence between the two models, implying that each model is expected to win on $D/2$ out of $D$ datasets. At the 5% significance level, a model is considered significantly better if it achieves at least $D/2 + 1.96\sqrt{D}/2$ wins. For RVFL, ELM, and BLS, the

Table 13: Nemenyi post hoc test results for **FLAiR**-enhanced RVFL, ELM, BLS, and dRVFL compared to their baseline models on binary and multiclass datasets.

| Model | Dataset type | Average rank | Rank difference from the baseline | Significant difference (As per Nemenyi post hoc test) |
|---|---|---|---|---|
| **FLAiR**-RVFL | Binary | 1.065 | 0.871 | Yes |
| | Multiclass | 1.009 | 0.979 | Yes |
| **FLAiR**-ELM | Binary | 1.048 | 0.904 | Yes |
| | Multiclass | 1.000 | 1.000 | Yes |
| **FLAiR**-BLS | Binary | 1.036 | 0.928 | Yes |
| | Multiclass | 1.000 | 1.000 | Yes |
| **FLAiR**-dRVFL | Binary | 1.026 | 0.968 | Yes |
| | Multiclass | 1.000 | 1.000 | Yes |

Table 14: Win-tie-loss test results comparing `StaR`-enhanced models with their baseline counterparts on binary and multiclass datasets.

| Model | Dataset type | Win | Tie | Loss |
|---|---|---|---|---|
| `StaR`-RVFL vs. RVFL | Binary | 65 | 23 | 5 |
| | Multiclass | 42 | 3 | 8 |
| `StaR`-ELM vs. ELM | Binary | 63 | 17 | 13 |
| | Multiclass | 44 | 3 | 6 |
| `StaR`-BLS vs. BLS | Binary | 68 | 18 | 7 |
| | Multiclass | 48 | 3 | 2 |
| `StaR`-dRVFL vs. dRVFL | Binary | 34 | 2 | 0 |
| | Multiclass | 32 | 0 | 3 |

Table 15: Win-tie-loss test results comparing `FLAiR`-enhanced models with their baseline counterparts on binary and multiclass datasets.

| Model | Dataset type | Win | Tie | Loss |
|---|---|---|---|---|
| `FLAiR`-RVFL vs. RVFL | Binary | 82 | 10 | 1 |
| | Multiclass | 52 | 1 | 0 |
| `FLAiR`-ELM vs. ELM | Binary | 85 | 7 | 1 |
| | Multiclass | 53 | 0 | 0 |
| `FLAiR`-BLS vs. BLS | Binary | 77 | 6 | 0 |
| | Multiclass | 53 | 0 | 0 |
| `FLAiR`-dRVFL vs. dRVFL | Binary | 33 | 3 | 0 |
| | Multiclass | 35 | 0 | 0 |

threshold for statistical significance is approximately 55.95 in the binary setting ($D = 93$) and 33.63 in the multiclass setting ($D = 53$). For dRVFL, the corresponding thresholds are 23.88 for binary ($D = 36$) and 23.30 for multiclass classification ($D = 35$). As evident from Tables 14 and 15, both `StaR` and `FLAiR` models consistently surpass their baseline counterparts by a statistically significant margin.

Taken together, these statistical evaluations offer robust empirical support for the effectiveness of the StaR and FLAiR frameworks. The consistent improvements across a wide range of datasets confirm that integrating `StaR` or `FLAiR` into RdNNs leads to significantly enhanced performance, both in terms of accuracy and stability. These results firmly establish the proposed frameworks as principled and effective enhancements for RdNNs.

## F CONVERGENCE BEHAVIOR OF `FLAiR`: EMPIRICAL VALIDATION OF MONOTONIC LOSS REDUCTION

To validate the theoretical guarantees of the `FLAiR` framework—particularly Lemma 3, which posits that gradient-based optimization of the input-to-hidden projections consistently reduces prediction error during the warm-up phase, we visualize the evolution of prediction loss over warm-up epochs. Specifically, we aim to empirically demonstrate that `FLAiR` yields a monotonic decrease in loss as the number of warm-up epochs increases. This analysis is conducted using the `FLAiR`-RVFL architecture, which integrates the `FLAiR` mechanism into the standard RVFL model. We evaluate this phenomenon on four datasets—Breast Cancer, Echocardiogram, Statlog Heart, and Titanic—by plotting the prediction loss (measured as the mean squared error between logits and targets) against warm-up epochs ranging from 2 to 10. Experiments are conducted under the same hyperparameter search protocol used for main evaluation, ensuring fair and consistent comparisons. As shown in Figures 1a, 1b, 1c, and 1d, the loss curves for all datasets exhibit a clear and consistent downward trend across increasing warm-up epochs. These observations clearly align with our theoretical formulation: `FLAiR` introduces task-awareness into the random input-to-hidden transformations by optimizing the hidden representations to minimize prediction error. This process enhances the adaptability of RdNNs while preserving their simplicity and computational efficiency. The consistently

decreasing loss curves empirically demonstrate that the warm-up phase effectively drives the model toward a more informative and stable feature space.

Figure 1: Warm-up epoch vs. prediction loss for four benchmark datasets under the **FLAiR** framework. Consistent loss reduction empirically supports the guarantee stated in Lemma 3.

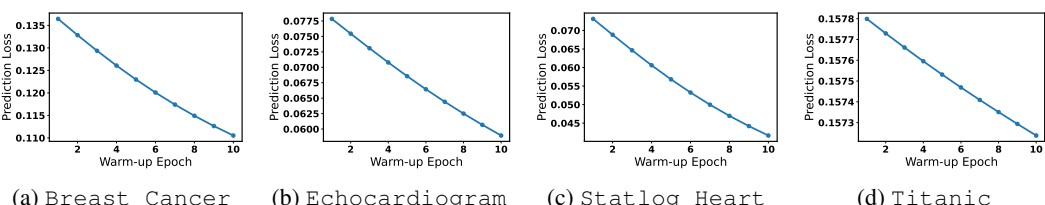

(a) Breast Cancer    (b) Echocardiogram    (c) Statlog Heart    (d) Titanic

