# OpenReview forum: "$\texttt{StaR}$ and $\texttt{FLAiR}$: Stabilizing and Enriching Randomized Neural Networks"
_ICLR.cc/2026/Conference — ICLR 2026 Conference Withdrawn Submission_

### Official Review · Reviewer_Mchm · 2025-10-19

**Soundness:** 3
**Presentation:** 3
**Contribution:** 2
**Rating:** 4
**Confidence:** 3

**Summary:**

The paper considers learning using randomized neural networks (RdNN). That is, networks where only the output layer is trained, and the other parameters are fixed at their random initialization. This approach has been studied in previous works, but here the authors suggest two improvements: (1) initializing the weight matrices such that all singular values are within a fixed range; and (2) adding a “warm-up” training phase, where the hidden parameters are updated for a small number of steps, and only then obtaining the output layer as in standard RdNN.

**Strengths:**

Understanding the power of random features / random networks is a natural question in deep learning. This approach has computational benefits over standard training (but also strong limitations). The current paper suggests two natural modifications for the naive approach, which somewhat improve the performance, and may be useful in practice.

**Weaknesses:**

The theoretical analysis in Appendix B is rather trivial, and therefore the main contributions of the paper are the new approach and the empirical results. I will now discuss it.

Let me start with FLAiR: Since the main limitation of RdNN is the absence of feature learning (as the feature map is fixed at initialization), it is natural to assume that the performance will improve if we add a feature-learning phase. If the feature learning phase is long enough, then we expect to obtain the performance of standard gradient descent, and if this phase is short, we expect to obtain performance which are better than standard RdNN but worse than standard gradient descent. Hence, I think that neither the approach nor the empirical results are particularly novel or surprising.

Regarding STaR: The approach of modifying the random initialization such that the singular values are bounded (and hence also the condition number) seems to be novel, as far as I am aware. If, for some reason, a practitioner wants to use RdNN, they might gain from using the modified initialization scheme suggested here. However, while the empirical results indicate that the performance gain is statistically meaningful, I would still argue that it’s rather modest. As can be seen in Table 1, the accuracy improvement is roughly 1% (and often less). Hence, this approach might be useful, but the performance gain is quite mild.

Overall, while the paper includes some useful ideas and observations, I think that the contributions are below the acceptance threshold.

**Questions:**

- May the authors elaborate on whether and when RdNN is used in practice?
- Is optimizing the output layer using a closed form expression that involves matrix inversion more computationaly efficient than performing SGD on the output layer? Even keeping the matrix H in memory might be difficult for large datasets.

---

### Official Review · Reviewer_rTXk · 2025-10-23

**Soundness:** 2
**Presentation:** 2
**Contribution:** 1
**Rating:** 2
**Confidence:** 3

**Summary:**

The paper proposes two methods to improve the performance of a two-layer randomized neural network, where the inner weight matrix W is randomly initialized and fixed during training. They proposed two methods:
1. do a linear interpolation to the singular values of W to make sure the singular values are within a pre-given range [$\sigma_{low}, \sigma_{high}$], so that W has good condition number.
2. first train W for a few steps, then freeze it in the remaining steps.

**Strengths:**

The writing is easy to follow. The proposed methods are reasonable and intuitive. The methods are validated on many tasks.

**Weaknesses:**

StaR is a rescale of the singular values to make W well-conditioned, which should be the first thing people think of when they encounter unstable training.

FLAiR treats the two-layer randomized neural network as a standard two-layer neural network in the first few steps. The idea is interesting, but the rule of when to stop training W is missing. It would be great if you can show
1. The total FLOPs needed of each method to achieve the same performance (1. W is always frozen, 2. W is always trainable, 3. W is first trainable, then becomes frozen).
2. For FLAiR, the total FLOPs needed to achieve the same performance for different warm-up steps.

Without these experiments, it's hard to say your method is better than models with trainable W, or is more efficient than models with W always frozen.

**Questions:**

Are RdNN architectures used in any realistic tasks? I don't know any LLMs using randomized weights/input embedding.

---

### Official Review · Reviewer_RF6M · 2025-10-27

**Soundness:** 2
**Presentation:** 2
**Contribution:** 2
**Rating:** 2
**Confidence:** 3

**Summary:**

This paper proposes two methods to improve Randomized neural networks:
- STaR improves the conditioning of the randomly sampled hidden layer weights.
- FLAiR encompasses a data-driven approach for initialization with several gradient updates to the hidden layer weights.
The authors showed that both methods improve the performance of randomized neural networks.

**Strengths:**

Experiment results show consistent improvements over the baseline. There are some interesting theoretical analyses.

**Weaknesses:**

Some aspects of the proposed method are not sufficiently motivated and ablated. For example:
- For StaR, why should we select linear rescaling among various methods to improve the conditioning? Other methods include simple clipping on extreme singular values, scaling on a logarithmic scale, and many more.
- The authors proposed to use a short warm-up phase for FLAiR, but I could not find discussions on how to select the length of this phase. The experiments (Figure 1) show that model performance improves monotonically with the number of warm-up epochs up to 10 epochs. How about beyond 10 epochs? Why do we need to keep the warm-up phase "short"? Can we take a fully pre-trained model like a ResNet and update the final layer weights with the closed-form solution from Eq. (2)?
- The authors claim that StaR and FLAiR preserve the efficiency of RdNNs, but this claim is not fully backed by experimental results. In particular, FLAiR seems to increase training cost. It is also unclear whether the proposed methods aare more efficient than traditional back-propagation when the number of hidden layers is large, which is a prerequisite of RdNN's universal approximation capability but cubically increase the computation for the matrix inversion in Eq. (2).

**Questions:**

- The square loss is often not the preferred loss function for classification. Can the proposed method extend beyond the square loss? Can it extend to non-classification tasks that use the square loss, such as but not limited to, regression and diffusion-based generation?

- The baseline numbers in Tables 1 and 2 seem different. Is there a reason behind this discrepancy?

---

### Official Review · Reviewer_DcXX · 2025-10-31

**Soundness:** 1
**Presentation:** 1
**Contribution:** 1
**Rating:** 2
**Confidence:** 4

**Summary:**

This paper proposes two complementary frameworks—STAR (Stabilizing Randomization) and FLAIR (Feature Layer Augmentation and Information Regularization)—aimed at enhancing the stability and expressive power of randomized neural networks (RNNets). Unlike conventional deep networks with learned weights, RNNets rely on random projections followed by closed-form optimization, which are fast but can suffer from instability and limited representational richness. The authors introduce (1) a stabilization technique that adaptively regularizes the random features to improve convergence and generalization, and (2) a feature enrichment strategy that augments the random mapping with structured representations to capture more complex dependencies.

**Strengths:**

The motivation to study in-depth randomized neural networks is clear.

**Weaknesses:**

1. While stability analysis is included, the theoretical part could better emphasize why STAR and FLAIR yield improved generalization (e.g., from the perspective of spectral bias or hypothesis class contraction).
2. It is not entirely clear how STAR and FLAIR interact—whether they are additive, complementary, or have diminishing returns when combined. A more detailed ablation study could clarify this relationship.
3. The overall contribution and quality of the paper fall short of the standard expected for an ICLR publication. The proposed randomized method does not represent a significant advancement in this research area, and the work lacks strong theoretical analysis to support claims of stability and generalization. Moreover, the experimental evaluation is not sufficiently compelling, as it does not include real-world applications or advanced benchmark tasks. The datasets used are outdated and have limited relevance to modern deep learning scenarios, making it difficult to convincingly demonstrate the true effectiveness and practical value of the proposed approach.
4. It appears that the authors have employed the controversial concept of “Extreme Learning Machine (ELM),” which has been the subject of considerable ethical discussion in the research community (see https://people.idsia.ch/~juergen/elmdispute1988-2007.html). Any further clarification?

**Questions:**

1. How sensitive are the results to the degree of randomization or initialization variance?

2. Could the stabilization mechanism be extended to other random-feature-based architectures (e.g., reservoir computing or extreme learning machines)?

3. Does FLAIR rely on domain-specific knowledge for feature augmentation, or is it fully data-driven?

4. How does the proposed framework behave when scaling to deeper randomized layers or high-dimensional inputs?

---

### Note · Authors · 2025-11-29

I have read and agree with the venue's withdrawal policy on behalf of myself and my co-authors.